# Dual-purpose dynamics emerge from a heterogeneous cell population in *Drosophila* metamorphosis

**Daiki Wakita** [1,2], **Satoshi Yamaji** [3,4], **Daiki Umetsu** [5]*, **Takeshi Kano** [6]*

**1** Misaki Marine Biological Station, The University of Tokyo, Miura, Kanagawa, Japan, **2** Japan Society for the Promotion of Science, Chiyoda, Tokyo, Japan, **3** Graduate School of Engineering, Tohoku University, Sendai, Miyagi, Japan, **4** Research Institute of Electrical Communication, Tohoku University, Sendai, Miyagi, Japan, **5** Graduate School of Science, The University of Osaka, Toyonaka, Osaka, Japan, **6** School of Systems Information Science, Future University Hakodate, Hakodate, Hokkaido, Japan

* tkano@fun.ac.jp (TK); daiki.umetsu.sci@osaka-u.ac.jp (DU)

**Data availability statement:** All relevant data are within the manuscript and its Supporting information files.

## Abstract

Collective cell behavior is fundamental to structure multicellular organisms. There, constituting cells always have heterogeneous properties across cell types (macro-heterogeneity) and within a cell type (micro-heterogeneity). Many studies have described the existence of heterogeneity in various systems at the cellular level, whereas little has investigated its effect on the systems. Unveiling how heterogeneity plays a role in the collective behavior would build a bridge from microscopic to macroscopic biological phenomena. We use the muscle remodeling in *Drosophila* as a model system, in which drastic reconstruction processes involve the physical interactions of various cells. At the early pupal stages, hundreds of hemocytes vigorously migrate and decompose the larval muscles into fragments called sarcolytes. We used in vivo and in silico approaches to understand how the dynamics of the sarcolyte population are characterized by hemocytes and other cells in the environment. Our in vivo imaging showed that the sarcolyte population gradually reduced the speed while changing the spatial arrangements. We also found that the sarcolyte dynamics involve macro-heterogeneity, namely, the coexistence of fat body cells and hemocytes, as well as micro-heterogeneity within hemocytes. To computationally evaluate the effects of factors determining the speed and arrangement of the sarcolyte population, we built a mathematical model assuming simplified interactions between sarcolytes, hemocytes, and fat body cells. Our simulations showed that, firstly, the efficient spreading and stable pattern formation of sarcolytes were together achieved by a delayed emergence of fat body cells and the micro-heterogeneity in hemocyte motility. Secondly, based on the similarity of observed and simulated network-like arrangements of sarcolytes, spatial confinement was another factor that causes the stabilization of sarcolytes. This study provides a pattern formation mechanism by which macro- and micro-heterogeneous migratory cells generate a 'dual-purpose' collective behavior—quickly spreading particles throughout the field while efficiently organizing them into an orderly arrangement.

**Funding:** This work was supported by JSPS KAKENHI JP21H05104 to TK, JSPS KAKENHI JP21H05105, JP24K02023, JP25H01363 to DU, JST FOREST Program J210000474 to DU, and Takeda Science Foundation to DU. DW received a salary from JP21H05104. The funders had no role in study design, data collection and analysis, decision to publish, or preparation of the manuscript.

**Competing interests:** The authors have declared that no competing interests exist.

## Author summary

Animals are made up of many different types of cells. Even in the same type, cells vary in shape and motion. How does this diversity affect the mass behavior of cells? We probed into how muscles change in shape inside a fly pupa, where many cells work together to reform the body. For example, 'hemocytes' move around and break the larval muscles into pieces. In a living pupa, we found that (1) the population of muscle pieces gradually slowed down and changed the patterns, (2) huge cells named 'fat body cells' were in the holes of the network-like patterns, and (3) hemocytes moved around in a cell-to-cell manner. We then used mathematical tools to find what affects the movement of muscle pieces. In computer simulations, we found that the factors to quickly spread muscle pieces while slowing them down into the pattern are (i) a later emergence of fat body cells, (ii) cell-to-cell variation in hemocyte movement, and (iii) a gradual narrowing of the space available for muscle pieces. We show how cell diversity achieves two purposes in parallel—scattering particles fast while settling them into a pattern.

## 1. Introduction

A number of individuals or cells often form a certain arrangement in nature. A flock of birds [1] and a school of fish [2] are examples that animal individuals form a swarm as if it is a huge migrating individual. Microscopically, each animal individual is structured by a collective of numerous cells. Similarly to animals in a population, cells in an animal are capable of migration, as represented by gastrulating cells in embryogenesis [3], epidermal cells in wound healing [4], and neutrophils in inflammation and infection [5]. One fundamental nature of multicellular systems is heterogeneity. For example, the human body includes various cell types such as epithelial cells, fat cells, muscle cells, and neurons [6]. Heterogeneity is also found within an apparently homogeneous cell population, as reported in the size and shape of red blood cells [7], the motility of sperm [8], and the gene expressions of embryonic stem cells [9] and of wound fibroblasts [10]. These cell-to-cell variations at different levels are comparable to 'macro-heterogeneity' (heterogeneity across cell types) and 'micro-heterogeneity' (heterogeneity within a cell type) [11]. While the existence of macro- and micro-heterogeneity has been described in various biological systems, its impact on collective cell behavior at the scale of systems has been less investigated. To understand how microscopic cells build up macroscopic morphogenesis at the tissue, organ, and individual levels, we need to take steps to understand how macro- and micro-heterogeneity in constituting cells characterizes the collective dynamics.

In holometabolous insects, the pupa is an exhibition where a huge number of various types of cells cooperatively behave to transform the body, thus providing a set of phenomena suitable to address the above issue. One representative phenomenon is muscle remodeling. At the early stage, the larval muscles are disassembled into numerous muscle fragments called 'sarcolytes' [12,13]. This fragmentation is mediated by the phagocytosis of hemocytes (also called plasmatocytes), which migrate with sarcolytes and exhibit micro-heterogeneity in motility such as velocity and directionality [12,14]. The dynamics of the sarcolyte population would be determined by the carrier hemocytes as well as other cells existing in the environment. However, the principal determinants remain to be elucidated.

Using *Drosophila melanogaster* at the early pupal stages, we aim to describe how the larval muscles are fragmented into sarcolytes in accordance with the surrounding macro- and micro-heterogeneous cells. Our primary focus is on their speed and spatial arrangement

because we hypothesize that scattered sarcolytes gradually settle in a fixed arrangement to facilitate the next process—possibly being digested or helping the formation of the adult organs. Our secondary focus is on the properties of the surrounding cells that are supposed to influence the movement of sarcolytes. Although rich genetic tools are available in *Drosophila*, in vivo approaches often face limitations in investigating a detailed mechanism because of the difficulty in eliminating a certain cell type and freely tuning cell properties while keeping other vital activities normal. One approach to overcome this issue is mathematical modeling with a small set of tunable parameters. The reliance of the modeled mechanism can be evaluated by quantitative comparisons between in vivo phenomena and in silico simulations.

Our results begin with the description of in vivo cellular dynamics—Sect 2.1. We found that the sarcolyte population gradually reduced the speed (Sect 2.1.1) while changing the network-like spatial arrangements (Sect 2.1.2). Observing the behavior of the surrounding cells, we found that dissociated fat body cells filled the void of sarcolytes (Sect 2.1.3) and that migrating hemocytes exhibited micro-heterogeneity in motile properties (Sect 2.1.4). These observations are followed by the computational results—Sect 2.2. Given the observed phenomena, we built a mathematical model assuming the counterparts of sarcolytes, hemocytes, and fat body cells with their contact-based interactions (Sect 2.2.1). Our simulations showed that, firstly, the quick spreading of sarcolytes and the efficient stabilization of their arrangement, namely, 'dual-purpose' dynamics, were realized by the later emergence of fat body cells (Sect 2.2.2) and the micro-heterogeneity in hemocyte motility (Sect 2.2.3). Secondly, based on the similarity of observed and simulated arrangements of sarcolytes, spatial confinement was another factor that causes the stabilization of sarcolytes (Sect 2.2.4).

## 2. Results

### 2.1. In vivo observations

Live imaging of the abdominal muscles in *D. melanogaster* pupae showed that the larval muscles decomposed into numerous sarcolytes, which moved in various directions while forming a network-like arrangement (Figs 1a and 2a). Of the swarm of sarcolytes, we quantified the extent of motion (Sect 2.1.1) and the spatial arrangement (Sect 2.1.2). We then visualized two types of the surrounding cells in vivo, fat body cells (Sect 2.1.3) and hemocytes (Sect 2.1.4), which are the candidates that characterize the dynamics of the sarcolyte population.

**2.1.1. Sarcolyte displacement becomes less prominent at later stages.** Sarcolytes persist up to 3–4 days after their generation upon histolysis [15] and move around in association with hemocytes [14]. To monitor the dynamics of sarcolyte movements, we began from investigating the time course of the sarcolyte displacement over a long time (>30 h) from 20 h after puparium formation (hAPF). We took images of sarcolytes in the abdomen of live pupae in which sarcolytes are labeled with a green fluorescent protein (GFP) fused to a muscle-specific protein, UNC-89, with a high time resolution (3 min) that allows moving individual sarcolytes to have overlapping pixels between two consecutive time frames. To better understand the dynamics of the sarcolyte movement, we sought to quantitatively analyze the velocity of sarcolyte movement in a systematic manner. We reasoned that to what extent signal intensity differs between time frames becomes a rough estimator for the average velocity of the sarcolyte population. For example, image sequences with static objects would show no difference in signal intensity, whereas those with displaced objects would show a large difference. Based on the original image sequence sampled every 3 min (Fig 1a), we obtained an image sequence of signal intensity difference (Fig 1b and S1 Video; algorithm shown in S1 Fig), from which we calculated $\Delta F(t)$, the spatial-average signal intensity of all pixels per frame (S2 Fig). However, the intensity signal of the entire image was changeable over time (Fig 1a), which would

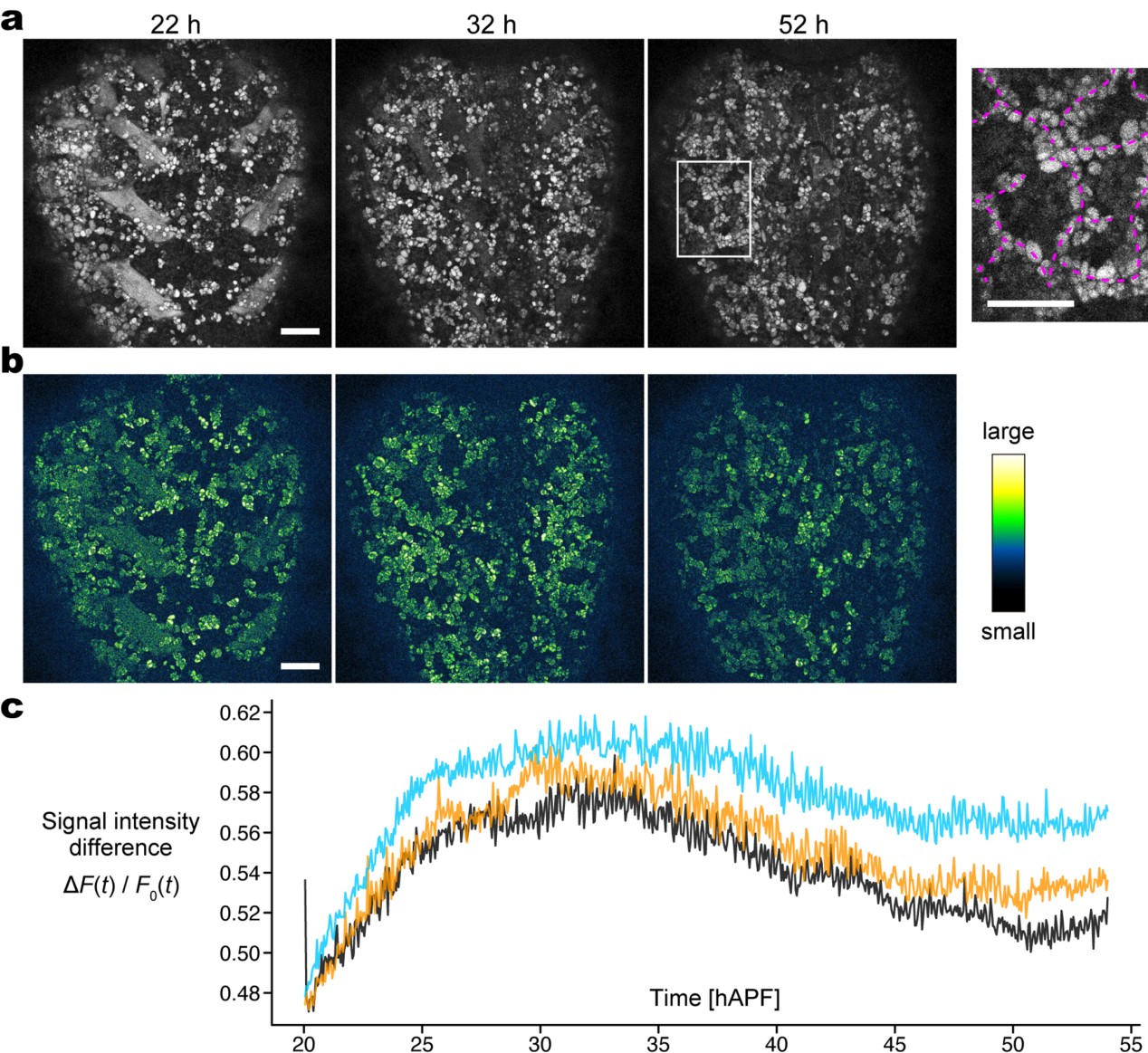

**Fig 1. In vivo sarcolytes (muscle fragments) slow down based on the signal intensity difference.** (a) Original snapshots at 22, 32, and 52 h after puparium formation (hAPF). The brightness is increased with a common degree to improve visibility. The rightmost inset shows a magnified view of the enclosed area at 52 hAPF and outlines one interpretation of a 'network-like' arrangement by the dashed curves. (b) Signal intensity difference images with the next frame (3 min later) at the snapshots in (a). While grayscale images were used for calculation, the displayed images are pseudo-colored to improve visibility: a large difference in bright yellow and a small difference in dark blue. Scale bars: 100 μm. (c) Normalized average signal intensity difference ($\Delta F(t)/F_0(t)$) in three individuals. The images in (a,b) were obtained from the individual of the black-colored plot. The spatial-average signal intensity of the original images ($F_0(t)$) and that of the signal intensity difference images before normalization ($\Delta F(t)$) are shown in S2 Fig. See S1 Video for the imaging video with the signal intensity difference in grayscale.

be a noise in quantifying the sarcolyte displacement. To reduce the noise, we divided $\Delta F(t)$ by $F_0(t)$, the spatial-average signal intensity of all pixels per frame from the original images (S2 Fig). We used the normalized sequence $\Delta F(t)/F_0(t)$, hereafter termed 'signal intensity difference,' to estimate the extent of motion of sarcolyte (Fig 1c).

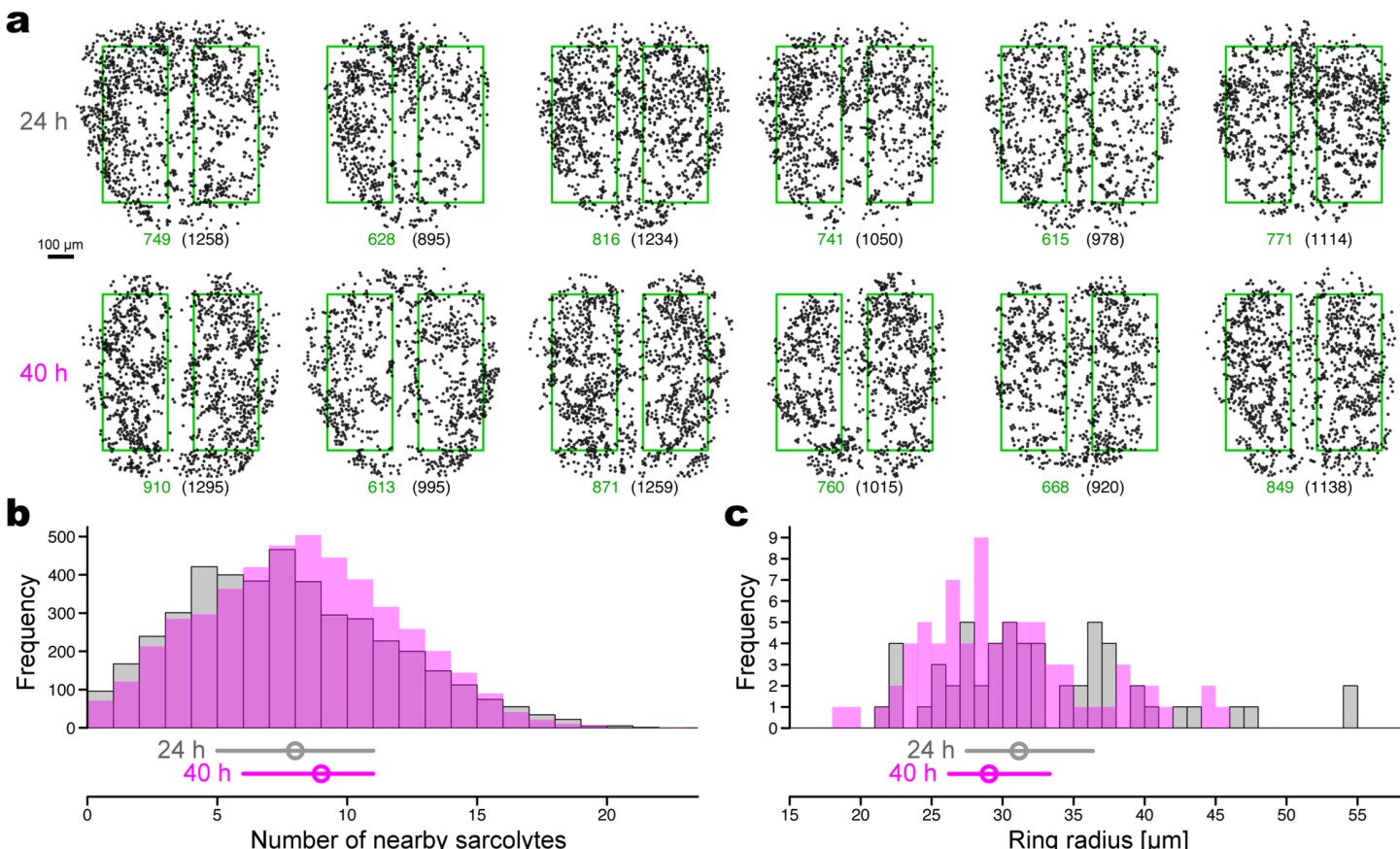

**Fig 2. Spatial arrangement of in vivo sarcolytes becomes fine and uniform from 24 to 40 hAPF.** (a) Two-dimensional coordinates of sarcolytes. Upper: 24 hAPF, separately showing six individuals. Lower: 40 hAPF, separately showing six individuals, different ones from those in the upper row. The inside of the green rectangles is analyzed to avoid marginal areas and the midline circulatory organ. The numbers below each coordinate data indicate the number of points inside the two rectangles (non-parenthesized) and the total number of points (parenthesized). (b) Histogram of the number of nearby sarcolytes, counted within 30-μm distance from each sarcolyte. Gray with outlines: 24 hAPF, magenta without outlines: 40 hAPF. The bottom plots represent the medians (circles) and interquartile ranges (error bars). Significant differences are found in rank average (Brunner-Munzel test, $p < 0.001$) and scatter (Siegel-Tukey test, $p < 0.001$). (c) Histogram of ring radius of sarcolytes, i.e., the death time of the 'ring-like' structures with >10 μm life time based on persistent homology. Plots are shown as in (b). Significant difference is found in rank average (Brunner-Munzel test, $p = 0.048$).

The signal intensity difference notably increased in 20–25 hAPF (Fig 1c). The peak at around 32 hAPF was followed by a gradual decrease to 50 hAPF ($n = 3$). The 1-h average around the minimum was 88.2%, 89.3%, and 93.1% of the 1-h average around the maximum for each pupa. If the signal intensity difference is linearly converted to the speed, we can interpret that the speed of sarcolytes decreased by ∼90% in 30–50 hAPF. However, signal intensity difference usually underestimates the original speed (S1 Fig). For example, when a sarcolyte has moved to a position at which another sarcolyte existed in the last frame, signal intensity around this position remains constantly high in some pixels, making signal intensity difference smaller than the extent of the actual displacement. Thus, the percentage in the actual speed is supposed to be lower than 90%. These data suggest that the sarcolyte movement has a certain temporal arrangement and eventually settles towards the late pupal stages. The mechanism of this slowdown is explored in silico in Sect 2.2.4.

**2.1.2. Sarcolyte network-like arrangement becomes fine and uniform.** Visual inspection gave us the impression that sarcolytes are arranged in a somewhat network-like arrangement (Fig 1a, right inset), which is a mesh- or web-like structure where multiple particles comprise its nodes and links. We therefore asked whether some orderly network-like arrangements emerge in the two-dimensional space as a consequence of vigorous movement followed by slow movement. To quantitatively investigate the feature of the sarcolyte arrangement, we tracked two-dimensional coordinates of sarcolytes at 24 and 40 hAPF (Fig 2a). Firstly, we counted the number of nearby sarcolytes for each sarcolyte within a distance of 30 μm. From 24 to 40 hAPF, the number of nearby sarcolytes significantly increased (Fig 2b; Brunner-Munzel test, $p < 0.001$). Medians for the number of nearby sarcolytes increased from eight to nine in this period, indicating that sarcolytes became slightly close to each other at the later stage. Moreover, the range of the number of nearby sarcolytes significantly decreased (Siegel-Tukey test, $p < 0.001$), indicating a more uniform distribution at the later stage.

Secondly, we sought to evaluate the geometrical pattern of the network-like arrangement of sarcolytes. One of the measures is the persistent homology, which describes the feature of voids in a point cloud based on the distances between points enclosing each void and the size of the void. Using HomCloud [16], we performed persistent homology to extract 'ring-like' structures, each of which represents a relatively large void enclosed by dense sarcolytes (algorithm detailed in Sect 4.3 and S3 Fig). A set of the radii of 'ring-like' structures, hereafter termed 'ring radius,' was used for the quantitative evaluation. Large values of ring radius represent a rough network-like arrangement, whereas small values of ring radius represent a fine network-like arrangement. We found that ring radius significantly decreased from 24 to 40 hAPF (Fig 2c; Brunner-Munzel test, $p = 0.048$), with medians 31.2 and 29.0 μm, respectively. This result indicates that the spaces devoid of sarcolytes became smaller in this period, with their radius around 30 μm. The range of ring radius was not significantly different between the examined time points (Siegel-Tukey test, $p = 0.46$). Together with the results of the number of nearby muscle units, the network-like profile of sarcolytes became fine and uniform in 24–40 hAPF.

**2.1.3. Fat body cells co-exist with sarcolytes.** A possible explanation for the presence of relatively regular void spaces with a radius of 30 μm for sarcolyte distribution is that there are some objects or cells occupying these voids. It has been reported that the fat body cells, which are gigantic cells about 30 μm in radius, migrate around in the body cavity of pupa in a similar time window as our observation [17]. To test if the fat body cells co-exist with sarcolytes and fill in the spaces devoid of sarcolytes, we fluorescently labeled fat body cells using a fat body GAL4 driver, Lsp2-GAL4, and performed live imaging.

During larval stages, fat body cells form a large sheet-like structure, fat body, in which constituent cells tightly adhere to each other [18,19]. Consistently with the previous studies, fat body cells dissociated around 15 hAPF (Fig 3a and S2 Video). As soon as the dissociation, fat body cells started migration and then intermingled with sarcolytes in a closely packed manner (Fig 3b). These observations clearly showed that the spaces devoid of sarcolytes were where fat body cells were present. The results suggest that apparent regularly spaced voids in the sarcolyte distribution within the abdomen reflect the relatively regular distribution of fat body cells embedded in the sarcolyte population. Since sarcolytes are engulfed in hemocytes, the co-existence with fat body cells thus corresponds to macro-heterogeneity in the collective system surrounding sarcolytes.

**2.1.4. Hemocyte population displays micro-heterogeneity in motion.** Sarcolytes are present inside hemocytes, which are phagocytic migratory immune cells in *Drosophila* [12].

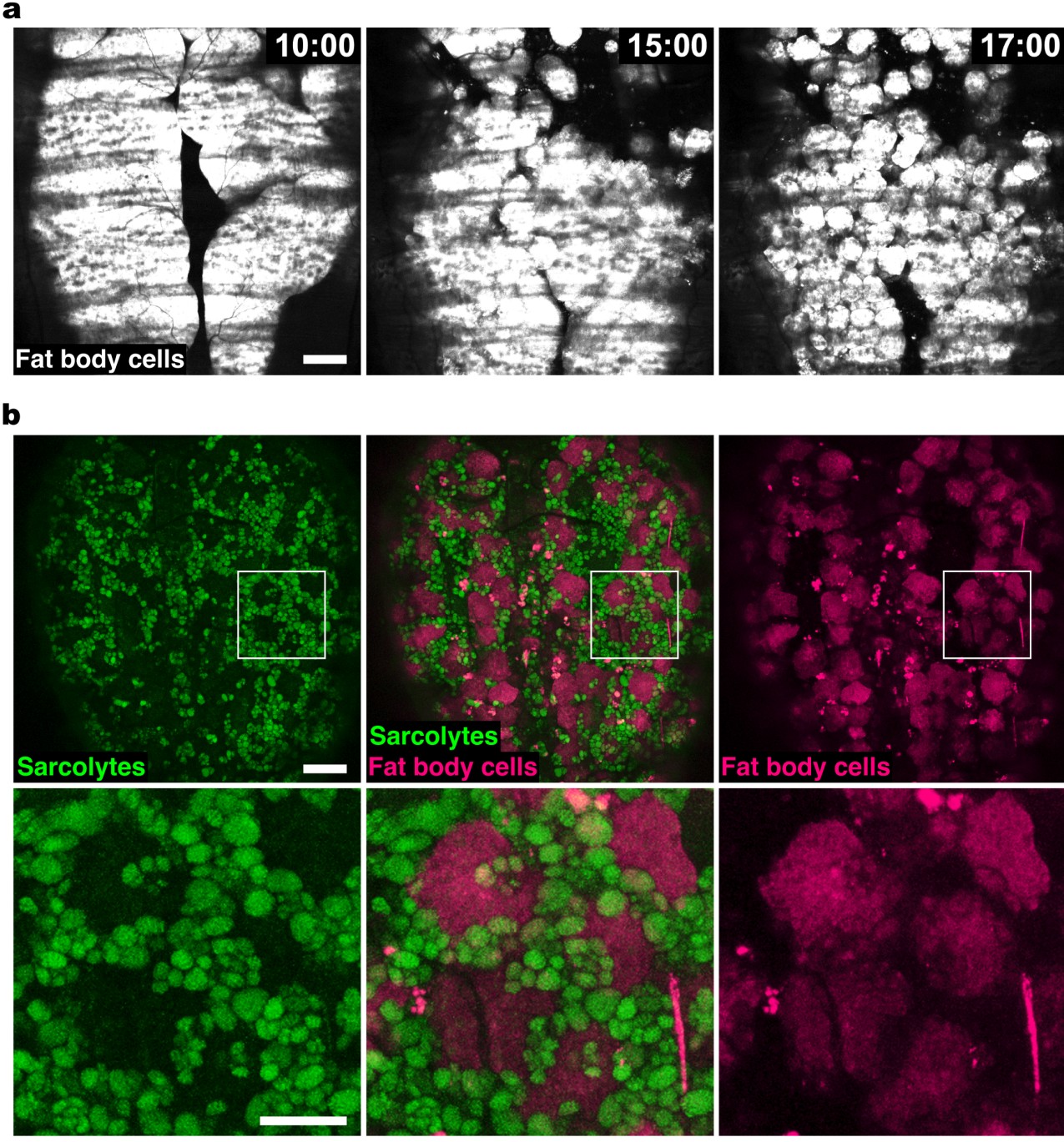

**Fig 3. In vivo fat body cells co-exist with sarcolytes in the pupal abdomen.** (a) Individualization of fat body cells during the early pupal stage. Fat body cells were visualized with expression of turboRFP (white) under the control of a fat body cell GAL4 driver, Lsp2-GAL4. Times after puparium formation (hAPF) are indicated at the top right corners (h:min). Scale bar: 100 μm. (b) Co-existence of fat body cells and sarcolytes. Sarcolytes (green) and fat body cells (magenta) were present in a nearly complementary arrangement at 43 hAPF. The magnified views of insets in the upper images are indicated in the lower images. Scale bars: 100 μm (upper images) and 50 μm (lower images). See S2 Video for the imaging video.

Moreover, the tracks of sarcolytes are well coordinated with those of hemocytes [14], indicating that the hemocytes provide sarcolytes with the driving force by holding phagocytosed sarcolytes during the pupal stage. We therefore turned our focus onto the analysis of hemocytes' motility, instead of sarcolytes, to understand the potential interaction between hemocytes and fat body cells and its contribution to the observed spatial network-like arrangement of sarcolytes in the tissue. To characterize the motion of hemocytes, we systematically quantified velocity and migratory direction consistency for hemocytes based on two-dimensional coordinates from 120 minutes tracking data for individual hemocytes (Fig 4).

Stacking of migration tracks of all analyzed hemocytes showed that the migration of overall hemocyte population is not unidirectional (Fig 4a; S5 Fig, a) with the variable migration speed within the population (S5 Fig, b). While the individual hemocytes moved in a relatively constant pace (Fig 4b), the velocity characteristic to individual cells was variable (Fig 4c), indicating that individual cells have their own typical migratory speed, and they are heterogeneous in migration speed.

Because migrating cells frequently change the migration directions due to the formation of unstable lamellipodia structures at their leading edge [20], we focused on how hemocytes

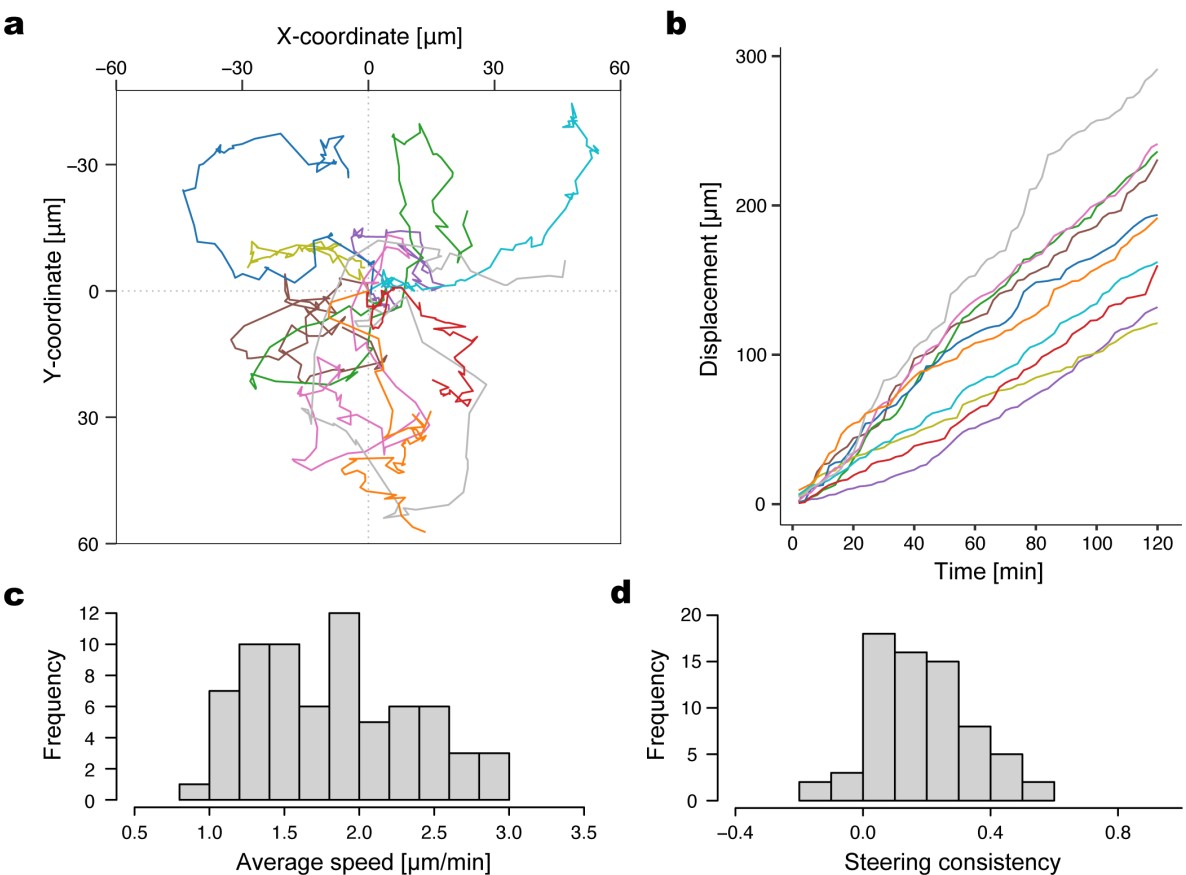

**Fig 4. In vivo hemocyte population exhibits micro-heterogeneity in speed and directional persistence.** (a) Trajectories of ten representative hemocytes. The initial positions are aligned to the origin. (b) Cumulative displacement of hemocytes. Colors correspond to the ten trajectories in (a). (c) Histogram of average speed of hemocytes. (d) Histogram of steering consistency, the degree to which each hemocyte maintains a consistent migration direction.

change the directions of their migration. We found that most hemocytes change the migration directions frequently (for a typical migration track, see S5 Fig, c). To analyze how much hemocytes change their migration directions, we measured angular difference of their migration directions between two consecutive time frames for individual cells (S5 Fig, c and d). Specifically, we defined 'steering consistency' of the $i$th hemocyte $S_i$ as

$$S_i = \frac{1}{N-2} \sum_{n=2}^{N-1} \cos\theta_i(n), \tag{1}$$

where $n$ denotes the time frame, and $N$ is the total number of the time frames; $\theta_i(n)$ denotes the angle between the velocity vector $\mathbf{v}_i(n)$ and $\mathbf{v}_i(n+1)$. Here, the velocity vector was calculated as $\mathbf{v}_i(n) = (\mathbf{r}_i(n) - \mathbf{r}_i(n-1))/\Delta t$ where $\mathbf{r}_i(n)$ is the position of the $i$th hemocyte at the $n$th time frame, and $\Delta t$ is the interval between the consecutive time frames. The steering consistency becomes closer to one when a hemocyte moves straight. Interestingly, like the migration speed, the steering consistency also exhibited heterogeneity among the hemocyte population (Fig 4d). These data indicate that the motion of hemocytes in both velocity and turning frequency displays remarkable micro-heterogeneity during the muscle remodeling in *Drosophila* metamorphosis.

## 2.2. In silico simulations

**2.2.1. Model.** To identify how macro-heterogeneity (existence of hemocytes and fat body cells) and micro-heterogeneity (variation in hemocyte motility) dictate the dynamics of the sarcolyte population, we introduced an in silico approach, which enables us to easily manipulate the existence and parameters of the cells. We built a mathematical model of three types of circular particles in two dimensions, that is, muscle units, hemocytes, and fat body cells, considering their friction and interactions via contact. The model is detailed in Sect 4.4.

- **Muscle unit**. We modeled the larval muscles, regardless of pre- or post-fragmentation, as circular particles hereafter termed 'muscle units' (Fig 5a). A muscle unit is stationary unless it contacts other self-propelling particles (Fig 5b). Each muscle unit has two states, 'uneaten' and 'eaten,' with the same shape but with different properties in interactions with other particles. Muscle units initially comprise the larval muscle before fragmentation in the uneaten state. Each muscle unit changes the states from uneaten to eaten through phagocytosis by a hemocyte (Fig 5c), as explained below. The eaten muscle unit thus represents a single sarcolyte after fragmentation.

- **Hemocyte.** For the carrier of muscle units, we modeled a hemocyte, a small particle that actively migrates with a strength of self-propelling force in random directions (Fig 5a). The moving direction is updated per step with a certain probability. To abstract phagocytosis, an enormous adhesion force is generated once a hemocyte contacts an uneaten muscle unit (Fig 5c). The hemocyte afterward becomes the host of the muscle unit; in other words, the muscle unit turns eaten and passively follows the migration of the host hemocyte. The eaten muscle unit is supposed to be enclosed by the cytoplasm of its host hemocyte (Fig 5c, dotted curve) and thus is no longer eaten by other hemocytes. In contrast, the hemocyte is capable of eating multiple muscle units.

- **Fat body cell.** Considering a large particle that may underlie the network-like arrangement of muscle units, we modeled a fat body cell with a large radius (Fig 5a). A fat body cell actively migrates with a strength of self-propelling force in random directions, with

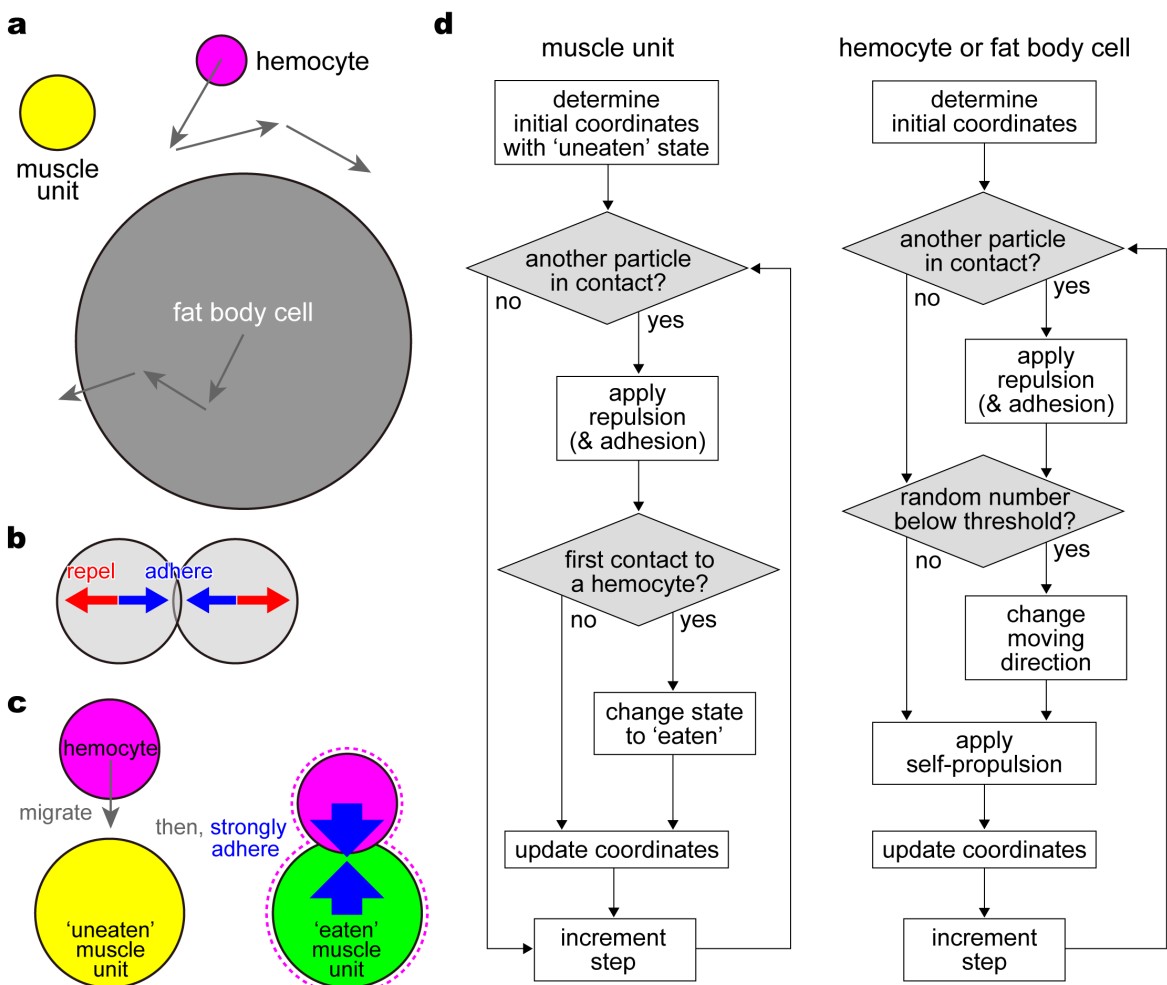

**Fig 5. Graphical abstract of the proposed agent-based model.** (a) Three types of circular particles assumed. A muscle unit does not actively migrate. A hemocyte and a fat body cell migrate with a speed ($v_i$ in Eq 3) and turn in a random direction with a probability ($p_i$). A fat body cell is much larger than the other two. (b) Interaction forces on two particles in contact. Repulsion force between any type of particles acts proportionally to the overlap length (Eq 4). Adhesion force between a certain combination of particles (Eq 5) acts in the direction opposite to the repulsion. (c) Phagocytosis. When a muscle unit in the 'uneaten' state (yellow) firstly contacts a hemocyte, the muscle unit is united with the host hemocyte by an enormous adhesion strength ($a_E$ in Eq 5) to become 'eaten.' Because the host hemocyte encloses the muscle unit in the cytoplasm as depicted by the dotted curve, the eaten muscle unit exhibits the hemocyte-like property in intercellular adhesion ($a_{HF}$ and $a_{HH}$ in Eq 5). (d) Simulation flowcharts for a muscle unit (left) and for a hemocyte or a fat body cell (right).

the moving direction updated as in hemocytes. Depending on the simulation conditions, each fat body cell gradually increases the radius from zero to its maximum, abstracting the floating process from the depth (Sect 2.1.3).

- **Friction and contact-based force.** Above-mentioned particles are subjected to friction from the environment (extracellular matrix) with a common friction coefficient, without inertia. Moreover, repulsion and adhesion forces are generated between two particles in contact (Fig 5b). Repulsion makes any type of particles (muscle unit, hemocyte, or fat body cell) move apart from each other, representing the excluded volume effect. Adhesion makes a certain combination of particles get close to each other. We assumed a certain strength of intercellular adhesion depending on simulation conditions.

Phagocytosis is implemented as adhesion between a hemocyte and a muscle unit, which is much stronger than the intercellular adhesion (Fig 5c). An eaten muscle unit, which is supposed to be inside a hemocyte, is treated as a hemocyte for intercellular adhesion but does not form phagocytosis adhesion with uneaten muscle units.

Under this agent-based model, the coordinates and state of each particle were updated every step according to contact-based and probabilistic decisions (Fig 5d). The process in which hemocytes disassemble the 'uneaten' larval muscles into 'eaten' sarcolytes is simulated in the following sections. We firstly performed simulations without and with fat body cells to highlight the effect of macro-heterogeneity on the dynamics of the sarcolyte population (Sect 2.2.2). To further examine if in vivo micro-heterogeneity of hemocytes observed in Sect 2.1.4 contributes to the disassembly and arrangement of the muscles, we compared simulations between homogeneous and heterogeneous hemocytes (Sect 2.2.3). We finally explored the mechanism of stabilization in sarcolyte displacement observed in Sect 2.1.1 by comparing several simulations that result in the slowdown of the muscle units via different ways (Sect 2.2.4).

**2.2.2. Macro-heterogeneity with hemocytes and fat body cells is essential to spread and arrange sarcolytes.** We firstly performed simulations under a periodic boundary condition with two types of particles, muscle units and hemocytes, to confirm if modeled hemocytes were able to disassemble the larval muscles consist of modeled muscle units. At the initial step of simulations, muscle units were randomly distributed within two belts, each representing the uneaten larval muscle (Fig 6a, 12 h), whereas hemocytes were distributed in the areas outside the belts. The muscle units were afterward carried by migrating hemocytes that occasionally contact with the belts (Fig 6a, 15 h), via the enormous adhesion representing phagocytosis (Fig 5c). Consequently, the belt-shaped larval muscles were fragmented to numerous particles, which eventually spread throughout the simulation field (Fig 6a, 24 h). The number of muscle units eaten by a single hemocyte was four at the maximum, which was determined mechanistically by the radii of hemocytes and muscle units.

To investigate the effect of fat body cells on the decomposition of the larval muscles, i.e., the effect of macro-heterogeneity, we compared the simulations without and with fat body cells. Without fat body cells, spreading muscle units remained in flux without forming a steady network-like arrangement after hours (Fig 6a, 24 h; S3 Video). We supposed that the large bodies of fat body cells would reduce the fluidity by filling the voids of the network-like arrangement as observed in vivo (Sect 2.1.3, Fig 3). When we distributed fat body cells around the larval muscles from the initial step, however, there was limited space for the muscles to scatter (Fig 6b). Throughout the simulation period, the average speed of muscle units was lower than that without fat body cells (Fig 6d, dashed curve). We also quantified to what extent muscle units spread over the field. Since the initial larval muscles were vertically arranged in the left and right sides, the degree of spreading is informed by the variance of the horizontal coordinate of muscle units in each side. The 'horizontal variance' mentioned hereafter is the average of the x-axis variances from the left and right halves of the field at each time point. In the presence of fat body cells, the horizontal variance was lower than that in the absence (Fig 6e, dashed curve), indicating a delay in spreading because of fat body cells. The speed and variance results together illustrate that fat body cells have an inhibitory effect on the decomposition process of the larval muscles while keeping muscle units slow throughout the time.

Facing this shortcoming, we considered the dissociation of fat body followed by the floating of fat body cells observed around 15 hAPF in vivo (Sect 2.1.3, Fig 3). In the simulations, the floating from the depth was implemented by the gradual increase in the radius of each

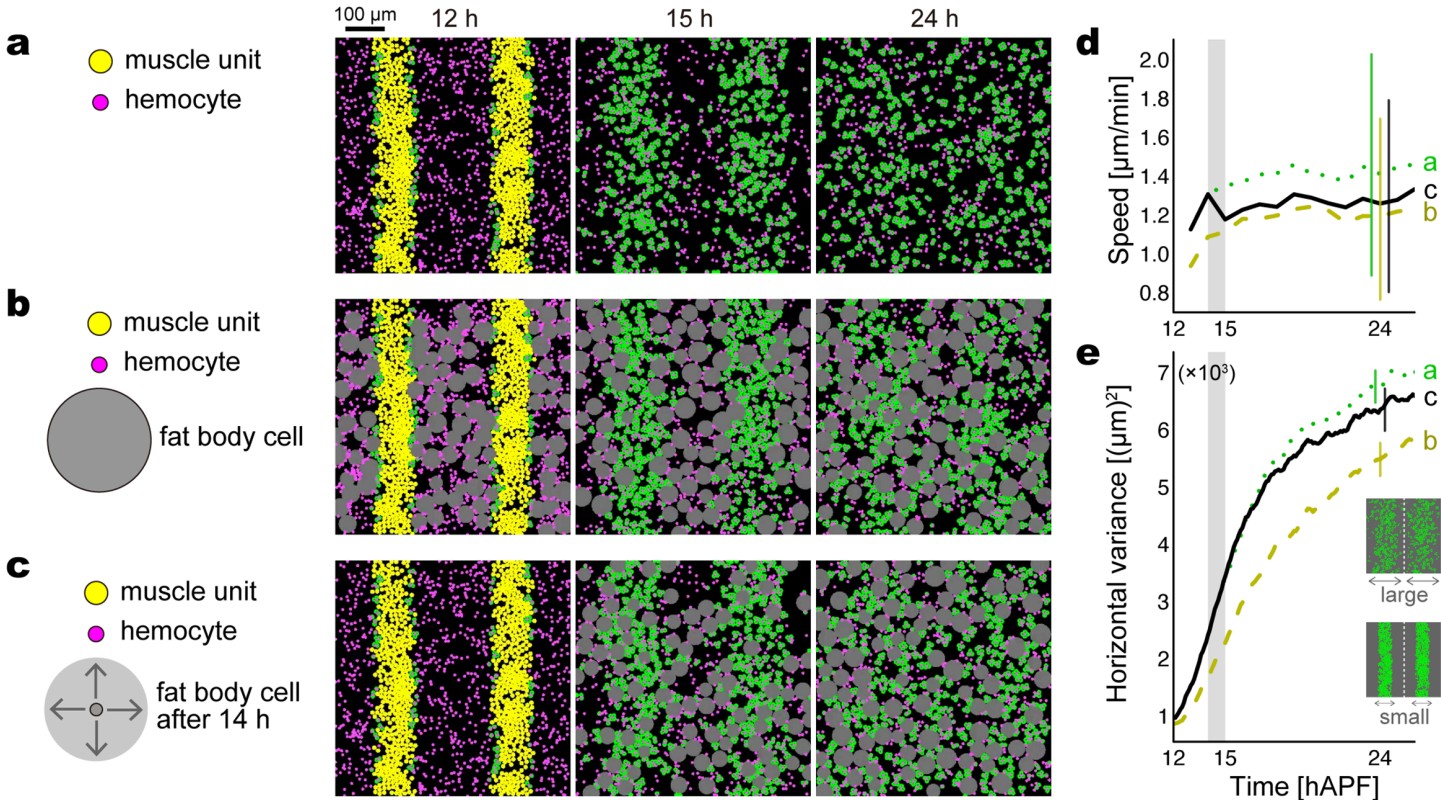

**Fig 6. Simulations highlight the roles of macro-heterogeneity (existence of hemocytes and fat body cells) in the sarcolyte dispersion and stability.** Muscle units and hemocytes exist in (a–c) with the same conditions. (a) Without fat body cells. (b) With fat body cells from the initial step, 12 hAPF. (c) With fat body cells that emerge after 14 hAPF. Snapshots in (a–c) are taken at 12, 15, and 24 hAPF (steps 240, 21,600, and 86,400, respectively) in a trial of each condition. In the simulation field, the yellow particles represent uneaten muscle units, initially distributing within two belts that imitate the larval muscles at 12 hAPF. The magenta particles outside the belts represent hemocytes, which actively migrate and engulf muscle units by an enormous adhesion force. The color of engulfed muscle units (sarcolytes) turns from yellow to green. Fat body cells colored gray exist from the beginning in (b), whereas their radii are gradually increased from zero to the cell-by-cell maximums ($R_i$) after 14 hAPF in (c), imitating the floating of fat body cells from the depth. In all the conditions, hemocytes are homogeneous in motility. (d) Speed of muscle units. Curves show medians over time, and error bars represent interquartile ranges at 24 hAPF, based on six trials (each 900 muscle units) for each condition. (e) Horizontal variance, i.e., x-coordinate variance of muscle units, averaged from the left and right halves of the field, as illustrated in the right inset. Curves show means over time, and error bars represent standard deviations at 24 hAPF, based on six trials for each condition. In (d,e), the three conditions (a–c) are represented by dotted, dashed, and solid curves, respectively. The shade in 14–15 hAPF indicates the floating period of fat body cells in the condition (c). See S3 Video for the simulation video.

fat body cell from zero to the cell-by-cell maximum, starting at 14 hAPF and completing mostly within 1 h (Fig 6c; Fig 6d and 6e, gray vertical band). The initial positions were random throughout the field, but the timing to float was delayed in areas where uneaten muscle units remain. As a result, the floating operation thereafter slowed down the displacement of muscle units (Fig 6d, solid curve). Because muscle units had 2 h to spread over the field before the floating, they were allowed to quickly form a wide network-like arrangement with its voids filled by fat body cells (Fig 6c, 24 h; S3 Video). The horizontal variance of muscle units was the middle of the simulations without fat body cells and those with initially existing fat body cells (Fig 6e, solid curve). Additional simulations showed that the adhesion between a hemocyte and a fat body cell enhanced the spreading of muscle units with a similar average speed (S6 Fig). To summarize this section, the sufficient spreading and stable pattern formation of muscle units are together accomplished by the initial disassembly by hemocytes, the subsequent

emergence of fat body cells, and the contact-based interaction between the two cell types. These results emphasize a functional importance of macro-heterogeneity.

### 2.2.3. Hemocyte micro-heterogeneity in turning frequency facilitates muscle decomposition while maintaining stability.
So far, our simulations showed a role of macro-heterogeneity for the efficient spreading. We next investigated the effects of micro-heterogeneity in the motile properties of hemocytes, which was observed in tissue (Sect 2.1.4, Fig 4). In the simulation setup, we considered three simplified scenarios in hemocyte motility as below to comparatively evaluate the collective dynamics of muscle units (Fig 7, S4 Video).

- Homogeneous hemocytes (Fig 7a). All the hemocytes were the 'standard type' with a common motile property.
- Heterogeneous hemocytes in self-propelling force strength (Fig 7b, termed 'speed heterogeneity'). Half the number of hemocytes were the 'slow type' with a small strength, while the other half were the 'fast type' with a large strength. The average strength was identical to that in the homogeneous case.
- Heterogeneous hemocytes in turning frequency (Fig 7c, termed 'turn heterogeneity'). Half the number of hemocytes were the 'meander type' with a high frequency, while the other half were the 'straight type' with a low frequency. The average frequency was identical to that in the homogeneous case.

Simulations showed that the average speed of muscle units was highest in the speed-heterogeneous scenario, intermediate in the turn-heterogeneous scenario, and lowest in the homogeneous scenario (Fig 7d). The horizontal variance in the distribution of muscle units was largest in the turn-heterogeneous scenario, intermediate in the speed-heterogeneous scenario, and smallest in the homogeneous scenario (Fig 7e). Note the reverse relationship of speed versus turn heterogeneity in the two measurements. In the heterogeneous conditions, the majority of muscle units were engulfed by the fast or straight type, while the meander type occupied a larger fraction than the slow type with the tested parameters (Fig 7f). These results indicate that, firstly, heterogeneity caused each muscle unit to be carried faster and enhanced the spreading of the larval muscles principally because of the fast or straight type, compared with homogeneity even if the average properties were identical. Secondly, the turn heterogeneity was capable of a more efficient spreading of the larval muscles while keeping each muscle unit carried slowly, that is, more stable, compared with the speed heterogeneity.

### 2.2.4. Spatial confinement induces in vivo-like changes in sarcolyte arrangement.
We finally aim to identify the mechanism of the slowdown of sarcolytes observed in vivo, hereafter termed 'stabilization' (Sect 2.1.1, Fig 1). Through simulations, we conducted five types of stabilization operations that similarly result in a slowdown of muscle units (Fig 8 and S5 Video). To conclude what stabilization mechanism is likely, we assessed whether each operation reproduces in vivo-like arrangement changes as observed in Sect 2.1.2. Below are the operations.

- Confine the space of the simulation field (Fig 8a).
- Decrease the self-propelling force strength of hemocytes (Fig 8b).
- Increase the friction coefficient of all the particles (Fig 8c).
- Increase the adhesion strength between two hemocytes (Fig 8d).
- Increase the adhesion strength between a hemocyte and a fat body cell (Fig 8e).

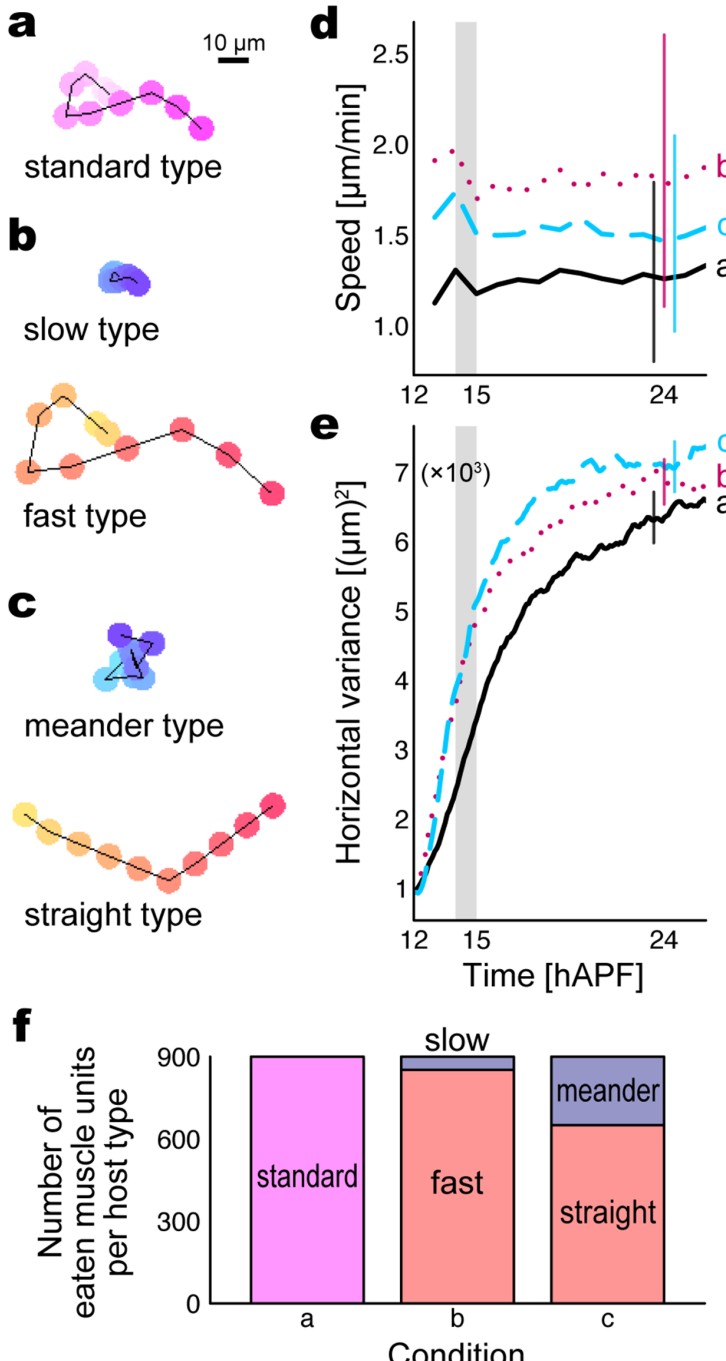

**Fig 7. Simulations reveal micro-heterogeneity in hemocyte motility contributes to the sarcolyte dispersion and stability.** (a–c) Simulated trajectories of representative hemocytes in three conditions: (a) homogeneity, with 800 hemocytes in standard type; (b) heterogeneity in the strength of self-propelling force, with 400 hemocytes in slow type and 400 hemocytes in fast type, termed 'speed heterogeneity'; (c) heterogeneity in turning frequency, with 400 hemocytes in meander type and 400 hemocytes in straight type, termed 'turn heterogeneity.' The color from light to dark indicates the time series sampled per minute (120 steps) for ten minutes. (d) Speed of muscle units. Curves show medians over time, and error bars represent interquartile ranges at 24 hAPF, based on six trials (each 900 muscle units) for each condition. (e) Horizontal variance (x-coordinate variance of muscle units, averaged from the left and right halves of the field). Curves show means over time, and error bars represent standard deviations at 24 hAPF, based on six trials for each condition. In (d,e), the three conditions (a–c) are represented by solid, dotted, and dashed curves, respectively. The shade in 14–15 hAPF indicates the floating period of fat body cells. (f) Average number of eaten muscle units sorted by the host hemocyte type at 24 hAPF (step 86,400) in the stacked bar chart. See S4 Video for the simulation video.

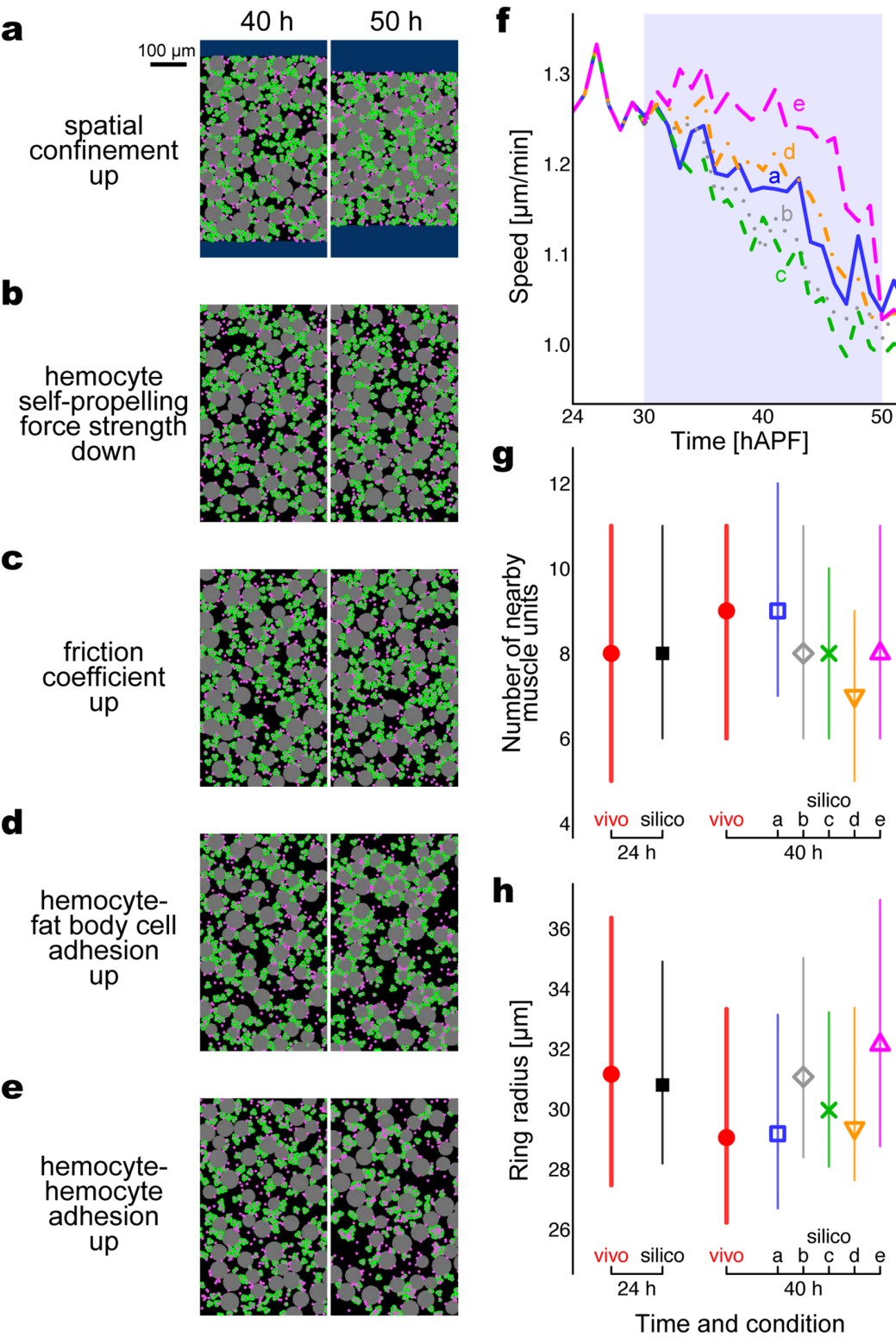

**Fig 8. Simulations suggest spatial confinement causes the late-stage rearrangement of sarcolytes.** Parameter changes to slow down the muscle units were performed from 30 to 50 hAPF (steps 129,600–273,600). (a–e) Snapshots of the left halves of simulation fields at 40 and 50 hAPF: (a) spatial confinement, (b) decrease in the self-propelling force strength of hemocytes, (c) increase in the friction coefficient of all the particles, (d) increase in the adhesion strength between a hemocyte and a fat body cell, (e) increase in the adhesion strength between hemocytes. In all the conditions, fat body cells started floating after 14 hAPF, and hemocytes were homogeneous in motility. (f–h) Evaluation plots. (f) Speed of muscle

units. Medians over time in the five conditions (a–e) are indicated by solid, dotted, short-dashed, dot-dashed, and long dashed curves, respectively. The shade indicates the stabilization period. (g) Number of nearby muscle units, counted within 30-μm distance from each muscle unit center. (h) Ring radius of muscle units based on persistent homology. Symbols and error bars in (g,h) show medians and interquartile ranges, respectively. The label "vivo" indicates the biological data shown in Fig 2b and 2c (filled circle). The label "silico" indicates the five conditions (a–e), with the common values before the operations (filled square) and diverged values during the operations (open square, diamond, cross, inverted triangle, and triangle, respectively). The spatial confinement simulations show similar changes to the biological data, namely, increase in the rank average of the number of nearby muscle units (Brunner-Munzel test, $p < 0.001$) and decrease in the rank average of the ring radius (Brunner-Munzel test, $p = 0.0038$). All the plots are based on six trials (each 900 muscle units) for each condition. See S5 Video for the simulation video.

The degree of parameter changes was determined to resultantly decrease the speed of muscle units by 80% between before and after the stabilization operation (30 and 50 hAPF, respectively; Fig 8f). Presuming that the decreasing rate of signal intensity difference from 32 to 50 hAPF in vivo, ∼90%, would underestimate the original speed (S1 Fig; explained in the last of Sect 2.1.1), we aimed a little lower criterion, 80%. We thus suppose that the decrease in speed in Fig 8f is comparable to the post-peak decrease in signal intensity difference in Fig 1c.

We evaluated to what extent the stabilization operations imitate the arrangement changes in vivo: increase in the average number of nearby muscle units, decrease in the range of the number of nearby muscle units, and decrease in average ring radius (Sect 2.1.2). Consequently, in terms of averages, the spatial confinement (Fig 8a) resulted in an arrangement change similar to that measured in vivo at 24 and 40 hAPF. That is, the number of nearby muscle units increased (Fig 8g, squares; Brunner-Munzel test, $p < 0.001$), indicating that they became closer to each other. Meanwhile, the decrease in ring radius (Fig 8h, squares; Brunner-Munzel test, $p = 0.0038$) represents a reduction in the void size of the network-like arrangement. The other four operations did not yield the two tendencies in average simultaneously. However, the interquartile range of the number of nearby muscle units, which reduced over time in vivo, was consistent between 24 and 40 hAPF through the in silico spatial confinement (Fig 8g, error bars around squares). These results indicate that arrangement changes in vivo are primarily caused by spatial confinement, while other factors may uniformize the arrangement to reduce the scattering of the number of nearby muscle units.

## 3. Discussion

In *Drosophila* metamorphosis, we described the in vivo dynamics of sarcolytes (muscle fragments, Figs 1 and 2) along with the two types of cells in the circumference, fat body cells (Fig 3) and hemocytes (Fig 4). Implementing the cellular dynamics into simulations, we suggest the advantages of macro-heterogeneity (existence of hemocytes and fat body cells, Sect 3.1) and of micro-heterogeneity (variation in hemocyte motility, Sect 3.2). The paper ends with a conclusion and outlook (Sect 3.3).

### 3.1. Advantage of macro-heterogeneity

In our simulations, hemocytes that come across the larval muscles (uneaten muscle units) fragment them into sarcolytes (eaten muscle units) and spread them throughout the space (Fig 6a). While sarcolytes appear to be incapable of active migration, their displacement is implemented via the attraction force between each hemocyte and muscle units as well as the self-propelling force of hemocytes. The attraction and self-propelling forces represent phagocytosis and active migration, respectively, as known from hemocytes in vivo [12,13]. No directional preference of the modeled hemocytes is required for the fragmentation and

spreading processes, which is compatible with a poor directionality in the actual hemocyte migration [12]. Another cell type, fat body cells (Fig 5a), was employed in the simulations given our in vivo observation that fat body cells filled the voids of the network-like arrangement of sarcolytes (Fig 3b). While hemocytes are an accelerator for spreading sarcolytes, fat body cells are a brake on the spreading because their large repulsive territories reduce the space available for sarcolytes (Fig 6b). Suppose this macro-heterogeneous collective system aims to achieve both the wide spreading and structural stabilization, the sarcolyte dynamics should be dominated by hemocytes at the early stage and by fat body cells at following stages. As notified by the dissociation of individual fat body cells from the continuum of the fat body after 15 hAPF in vivo (Fig 3a), the emergence of fat body cells after sufficiently spreading sarcolytes indeed facilitated the formation of a wide-spread, stable network-like arrangement with the voids filled (Fig 6c). The floating, a subsequent process after dissociation, is supposed to occur actively with the migration ability of fat body cells shown previously [17] or passively through the pupal transformation. These simulations provide a prediction that the induction of the immigration and phagocytosis of hemocytes should precede the induction of the dissociation of fat body cells. Based on molecular studies, the hemocyte activation may involve 20-hydroxyecdysone (ecdysone) signaling which triggers the degeneration of larval tissues including muscles during metamorphosis [21,22]. The fat body cell dissociation may also involve the ecdysone, which regulates the expression of matrix metalloproteinase, a requisite for fat body remodeling [18]. Given that both the events are possibly regulated by the same signaling, the time difference between them could occur by a difference in sensitivity to the ecdysone signal or a delay in the immigration of dissociated fat body cells into the muscle area.

Although the emergence of fat body cells works in gently keeping the sarcolyte arrangement at the early stages, their existence was not enough to achieve the gradual slowdown of the sarcolyte population after 30 hAPF (Fig 1c). Our further simulations suggested that spatial confinement after 30 hAPF is a primary stabilizing factor since the arrangement changes through this operation resembled those observed in vivo (Fig 8g and 8h). In the simulations, the slowdown mechanism by the confinement walls is identical to that by fat body cells. That is, the additional repulsive areas of the walls narrow the range of sarcolyte movement. In the actual pupa, the internal space may be confined by the adult organs that gradually become stiff during metamorphosis. In summary, this section suggests a step-by-step mechanism of muscle remodeling; the sarcolyte population firstly spreads via carriage by hemocytes, secondly forms a network-like arrangement via filling by fat body cells, and finally fixes the arrangement via confinement by other organs.

### 3.2. Advantage of micro-heterogeneity

Our in vivo quantitative analysis revealed micro-heterogeneity in the migration speed and turning frequency of hemocytes (Fig 4). This agrees with a previous study that shows a heterogeneity in the migratory behavior of hemocytes [12]. Our simulations then demonstrated that heterogeneous hemocytes in the self-propelling force strength and turning frequency both spread muscle units over the field more efficiently than homogeneous hemocytes (i.e., steeper increase in the horizontal variance, Fig 7e). The major hemocytes that engulfed sarcolytes belonged to the fast type in the speed heterogeneity and the straight type in the turn heterogeneity (Fig 7f). These are thus dominant types that characterize the dynamics of the sarcolyte populations. The fast and straight types traveled further than the slow and meander types, respectively (Fig 7b and 7c), having a higher frequency to collide with uneaten muscle units, i.e., to deconstruct the larval muscles. At the same time, the fast and straight types

also have frequent contact with hemocytes in the opposite types and thus enhance their displacement as well. This is a possible reason that the heterogeneous hemocyte populations have a greater capacity of spreading sarcolytes than the homogeneous counterpart with the same average in the parameter values. The speed of cell migration is positively correlated with the ratio of adenosine triphosphate (ATP, energy storage molecule) [23], which is required to modify the actin cytoskeleton [24] and thus to generate the self-propelling force of cells. The turning frequency would be inversely proportional to directional persistence, the duration in which a cell sustains a uni-directional migration. This duration partly depends on the duration of actin polymerization at the front of the cell [20], also requiring ATP. We thus speculate that the hemocyte populations may consume similar amounts of chemical energy when they have the same average level in the self-propelling force strength or the turning frequency. This suggests a biological advantage of micro-heterogeneity, under which the limited resource in a pupa enables a rapid decomposition of the larval tissue to promote the metamorphosis.

We next discuss the comparison between the two kinds of hemocyte heterogeneity described in vivo (Fig 4) and evaluated in silico (Fig 7). Our simulations showed that the turn heterogeneity resulted in a lower sarcolyte speed but a quicker spread than the speed heterogeneity (Fig 7d and 7e). The main reason for these apparently contradictory results is ascribed to the trajectories of the fast- and straight-type hemocytes, which dominate the displacement of sarcolytes as stated in the previous paragraph. Compared with the fast type, the straight type usually travels a shorter distance per minute (see the distances between consecutive points in Fig 7b and 7c) but a longer distance in a longer period (see the distances between the lightest and darkest points in Fig 7b and 7c). The short-term displacement is represented by the first measurement, the average speed (Fig 7d), which was originally calculated with short intervals (6 min, methods shown in Sect 4.6). Meanwhile, the long-term displacement would be reflected in the other measurement, how quickly the horizontal variance increases (Fig 7e). In the speed heterogeneity, the fast-type hemocytes rigorously travel in multiple directions in a short time, disturbing nearby sarcolytes and other hemocytes. In contrast, the straight-type hemocytes in the turn heterogeneity promptly penetrate the inside of the larval muscles and then linearly carry sarcolytes far away without disturbing the surroundings. Considering such a difference, we suppose that sarcolytes were carried by the dominant hemocytes with different distances depending on the time scale and thus showed the opposite relationship in the two measurements.

While we above focused on the fast and straight types as the major determinant of the sarcolyte dynamics, we also suggest the benefits from the existence of the slow and meander types. One possibility is that the limited moving range of these hemocytes (Fig 7b and 7c) serves a glue-like function to maintain the arrangement of the sarcolyte population. Particularly with the parameter of the turn heterogeneity, more than a quarter of the muscle units were engulfed by the meander-type hemocytes (Fig 7f), which can act as a brake. This may be another reason why the turn heterogeneity yielded a lower speed of muscle units than the speed heterogeneity (Fig 7d), under which muscle units eaten by the slow type were less than a tenth (Fig 7f).

Micro-heterogeneity in other cell types, such as fat body cells, may also provide advantages. In our simulations, round-shaped fat body cells floated simultaneously at a uniform rate throughout the field. However, further biological observations of the fat body dynamics could reveal micro-heterogeneity in dissociation and floating processes as well as cell morphology and motility. For example, a previous study reported the migratory properties of fat body cells during wound healing [17], in which we can recognize considerable variations in speed and meandering measurements. Moreover, such variations may be influenced by

internal environmental factors, including spatial constraints imposed by the pupal structure. Incorporating additional variability into simulations could uncover further roles of cellular micro-heterogeneity.

### 3.3. Conclusion and outlook

Our in vivo and in silico suggestions highlight the emergence of 'dual-purpose' dynamics from heterogeneity in cell properties such as the size, immigration stage, contact-based interaction, and motility. That is, the heterogeneity becomes a factor to achieve two different purposes during *Drosophila* muscle remodeling: quickly spreading particles over the field (speeding up the long-term displacement); meanwhile, efficiently organizing the particles into an orderly arrangement (slowing down the short-term displacement). Regardless of whether the heterogeneity is categorized into macro or micro, both levels of the heterogeneity contribute to this achievement. Other advantages of heterogeneity have been discussed in several cellular systems. One representative is the adaptability to multiple situations, as mentioned for the swimming phenotypes of chemotactic bacteria [25] and the exploratory behavior of slime molds [26]. Neuroscience studies have inferred, for example, the improvement of specific capacities by heterogeneous spiking thresholds [27] and the enrichment of color vision by spectrally heterogeneous photoreceptors [28]. Dual-purpose nature, the potential of one system to produce two types of outcomes, is a collective intelligence we newly emphasize in the heterogeneity studies. Further biological experiments are needed to validate the proposed benefits of macro- and micro-heterogeneity. Feedback from our in silico prediction to in vivo validation is, however, currently challenging due to the limited knowledge of molecular mechanisms regulating hemocyte micro-heterogeneity and fat body disintegration, thus awaiting thorough genetic studies in the future.

While we phenomenologically assumed that the sarcolyte population is designed to be scattered and patterned, the functional significance of forming a stable network-like arrangement in metamorphosis remains obscure. We here propose four hypotheses. First, because the fat body is a nutrient storage organ [29], we presume that fat body cells distributed in the voids of the network-like arrangement are not merely a spacer but a source of nutrition helping de- and reconstruction throughout the area. Second, the network-like arrangement may act as a scaffold to construct the adult organs. A bioengineering study showed that human cells on a non-aligned or large-pored collagen architecture (i.e., a rough network-like arrangement) proliferated more than those on an aligned or small-pored one [30]. Such a mechanistic preference may be involved even in totally different systems. Third, hemocytes lining the network may transfer some signals throughout the layer. This possibility arises from a report that hemocyte aggregates in insects form intercellular structural and electrical coupling via gap junctions [31]. Last, sarcolytes derived from the larval muscles may be reused for part of the adult muscles, with the network-like form being a precursor. In *Drosophila*, such reuse is known in the longitudinal visceral muscle cells of the larval gut, which dedifferentiate and then redifferentiate into adult muscle fibers [32]. By understanding the functionality of the pattern formation, we will see how the heterogeneity proposed in our model enhances the metamorphosis of holometabolous insects. Moreover, incorporated with a functional purpose, our model may be used to draft a swarm robot system that quickly and widely distributes some matters while maintaining their network connection. This aligns with a recent scheme of swarm robotics; heterogeneity with complementary skills plays a role in carrying out complex tasks in real-world scenarios [33].

## 4. Methods

### 4.1. Ethics statement

Ethical approval was not required for this study because insects are not subject to animal ethics regulations under current institutional and national guidelines.

### 4.2. Animals and live imaging

The following fly stocks were used: fTRG-Unc89 (GD318326, Vienna Drosophila Resource Center), Srp::Hemo-H2A.3XmCherry (BL78361, Bloomington Drosophila Stock Center), and Lsp2-GAL4 (BL6357, Bloomington Drosophila Stock Center) for visualizing sarcolytes, hemocytes, and fat body cells, respectively. Flies were raised at 25°C on standard food. Prepupae were collected and incubated at 25°C until imaging. For staged pupae, the pupal case was carefully removed on top of abdominal segments 1–4. The animals were placed on a coverslip, and images were acquired on a Zeiss LSM 900 inverted confocal microscope (x10 objective DRY lens, Zeiss Plan-APOCHROMAT 10x/0.45). Images were acquired at time intervals of 3 or 6 min.

### 4.3. In vivo evaluation

Image processing was performed to capture a temporal change in the extent of motion of sarcolytes. The signal intensity difference images between continuous frames were obtained using the "difference" operation in the image calculator of Fiji/ImageJ (NIH) (S1 Fig). The average signal intensity over all the pixels of the signal intensity difference images, $\Delta F(t)$, was calculated at each frame (S2 Fig). The value $\Delta F(t)$ was then divided by $F_0(t)$, the average signal intensity over all the pixels at the earlier original frame of the two continuous frames (S2 Fig). This normalization prevents that the signal intensity difference in interest is affected by the absolute signal intensity of the original images, which may vary over time by the experimental artifacts.

Two-dimensional coordinates of sarcolytes were used to capture a temporal change in spatial arrangement. Rectangle areas with width 250 μm and height 600 μm in the left and right, 100 μm apart from each other, were used for the later analyses to avoid unclear marginal areas and the midline circulatory organ (Fig 2a). In total, 24 rectangles were obtained from 12 individuals: six from 24 hAPF and other six from 48 hAPF. To quantify to what extent sarcolytes gather with each other, the number of sarcolyte points within a distance of 30 μm was counted for each sarcolyte point. In addition, to capture the feature of network-like arrangement, persistent homology was performed using HomCloud [16]. The algorithm is that, briefly, the radius of each sarcolyte point is gradually increased from zero at a constant rate, while a void enclosed by enlarging circles appears at a certain radius ('birth time') and disappears at a larger radius ('death time') (S3 Fig). The birth time is roughly equivalent to half the average inter-point distance enclosing the void, whereas the death time is roughly equivalent to the radius of the void. The output of persistent homology is a persistence diagram (S4 Fig), a scatter plot of the pairs of birth and death times. The subtraction of a birth time from the corresponding death time, termed the 'life time,' represents to what extent the group of points looks like a ring, which is usually characterized by not a few vertices enclosing a large void (S3 Fig). The death time whose life time was more than 10 μm was defined as the 'ring radius.' For both the number of nearby sarcolytes and the ring radius, the differences between 24 and 40 hAPF were statistically tested using the Brunner-Munzel test for the rank average and the Siegel-Tukey test for the scatter.

For measuring the migratory speed and direction consistency of hemocytes, two-dimensional coordinates of hemocytes labeled with Srp::Hemo-H2A.3XmCherry were used. The speed was measured from the live imaging data acquired every 2 min from 26 to 28 hAPF. Direction consistency was defined in Sect 2.1.4, termed 'steering consistency.' The dataset of each measurement is stored in S1 Dataset.

## 4.4. Modeling

To extract the significant properties of the relevant cells upon the collective dynamics of sarcolytes, we built a mathematical model assuming three types of circular particles: muscle units, hemocytes, and fat body cells (Fig 5a). A brief outline of the model was stated in Sect 2.2.1.

We suppose that the motion at the cell scale is dominated by frictional force and thus ignore inertia, as generally mentioned [34]. The two-dimensional central coordinates of particle $i$ (muscle unit, hemocyte, or fat body cell) at time $t$, denoted by $\mathbf{x}_i(t)$, are updated as

$$\mu \frac{\mathrm{d}\mathbf{x}_i(t)}{\mathrm{d}t} = v_i \mathbf{e}_i(t) + \sum_{j \in Z_i(t)} \mathbf{f}_{ij}(t), \tag{2}$$

where $\mu$ is a friction coefficient, $v_i$ determines the strength of the self-propelling force of the particle, and $\mathbf{e}_i(t)$ is a unit vector in the migration direction. The direction of $\mathbf{e}_i(t)$ is updated randomly with probability $p_i$ at each time step (cf. decision of "random number below threshold?" in Fig 5d). The self-propelling force strength $v_i$ depends on the type of particle $i$:

$$v_i = \begin{cases} 0 & \text{if } i \text{ is a muscle unit,} \\ v_{\mathrm{H},i} & \text{if } i \text{ is a hemocyte,} \\ v_{\mathrm{F}} & \text{if } i \text{ is a fat body cell,} \end{cases} \tag{3}$$

where we did not assume active migration of muscle units because no motor structure has been found. In contrast, based on biological findings, we assumed the active migratory ability of hemocytes [21] and fat body cells [17] (Fig 5a). The cell-by-cell index of $v_{\mathrm{H},i}$ is used to employ heterogeneity in the strength of the hemocyte self-propelling force for the later simulations.

The term $\sum_{j \in Z_i(t)} \mathbf{f}_{ij}(t)$ in Eq 2 represents a summation of contact-based interaction with particle $j$ that belongs to $Z_i(t)$, which is a set of any type of particles in contact with particle $i$ (cf. decision of "another particle in contact?" in Fig 5d). The force $\mathbf{f}_{ij}(t)$ consists of repulsion, a force to move away from each other, and adhesion, a force to get close to each other as below (Fig 5b).

$$\mathbf{f}_{ij}(t) = \left\{ -k(r_i + r_j - |\mathbf{x}_j(t) - \mathbf{x}_i(t)|) + a_{ij} \right\} \mathbf{e}_{ij}(t), \tag{4}$$

where the positive constant $k$ determines the strength of repulsion, $r_i$ and $r_j$ are the radii of particles $i$ and $j$, respectively, and $\mathbf{e}_{ij}(t)$ is a unit vector from $\mathbf{x}_i(t)$ to $\mathbf{x}_j(t)$. The expression $r_i + r_j - |\mathbf{x}_j(t) - \mathbf{x}_i(t)|$ represents the overlapping length of particles $i$ and $j$. Due to multiplication by the negative coefficient $-k$, the more the two particles overlap, the stronger the repulsion between them.

The strength of adhesion, which acts in the direction opposite to repulsion, is determined by the parameter $a_{ij} \geq 0$ in Eq 4. We define both of intercellular adhesion and phagocytosis

with different parameters:

$$a_{ij} = \begin{cases} a_{HF} & \text{between a hemocyte and a fat body cell (intercellular adhesion),} \\ a_{HH} & \text{between two hemocytes (intercellular adhesion),} \\ a_{FF} & \text{between two fat body cells (intercellular adhesion),} \\ a_E & \text{between a hemocyte and a muscle unit that has been uneaten} \\ & \text{(phagocytosis),} \\ 0 & \text{else,} \end{cases} \tag{5}$$

where $a_E$ should be much larger than the others and represents the strength by which a muscle unit is united with a hemocyte via phagocytosis. Once a hemocyte contacts a muscle unit that is in the 'uneaten' state at the moment before contact, the muscle unit becomes 'eaten' (cf. decision of "first contact to a hemocyte?" in Fig 5d). Because we suppose that the eaten muscle unit is enclosed by the cytoplasm of its host hemocyte, the eaten muscle unit is treated equally to the hemocyte upon the judgment of $a_{HF}$ and $a_{HH}$. For example, the inter-hemocyte adhesion strength $a_{HH}$ is also assigned between a hemocyte and an eaten muscle unit as well as between two eaten muscle units. The other properties of the muscle unit are not changed from the uneaten state; the eaten muscle unit remains incapable of active migration and incapable of engulfing another muscle unit to avoid a single hemocyte enclosing a limitless number of muscle units. Although the eaten muscle unit follows the consistent host hemocyte, the enormous adhesion by $a_E$ may be rarely disconnected by excessive repulsion from the surrounding particles. In this exceptional case, the eaten muscle unit returns to the uneaten state and can accept another host hemocyte. A hemocyte can be a host of multiple muscle units.

## 4.5. Simulation conditions

To reproduce the collective dynamics of modeled particles while imitating the spatial feature in vivo, we performed computer simulations with a parameter set under the mathematical model (Table 1, source codes available in S1 File).

**Parameter setting.** The radius parameters (muscle unit $r_S$, hemocyte $r_H$, and fat body cell $r_F$) were set according to biological knowledge [12,17] and our observations (Fig 3b). The friction coefficient $\mu$ and the self-propelling force strength of hemocytes $v_{H,i}$ were adjusted to yield a hemocyte speed of 1–3 μm/min, consistent with our experimental data (Fig 4c). The self-propelling force strength of fat body cells, which appeared to have a limited effect on the overall dynamics, was set to a small value. The turning probability $p_i$ was tuned to reproduce the observed trajectories of hemocytes [14]. The repulsion-related parameter $k$ was set to allow a partial overlap between cells given their deformable nature. Adhesion strength parameters ($a_{HF}$, $a_{HH}$, $a_{FF}$, and $a_E$) were tuned to reproduce a network-like arrangement of muscle units while avoiding unnatural clustering (S6 Fig).

**Time evolution and field.** The differential equation (Eq 2) was solved using the Euler method. The time interval per step, d$t$, was set to 0.5 s with the maximum step 295,200, corresponding to 41 h. We assumed that each simulation begins at 12 hAPF, from which muscle units were observed in vivo; thus, the ending time corresponds to 53 hAPF. The simulation area was a square of $600 \times 600$ μm with a periodic boundary condition applied to the four sides. The modeled particles were distributed in this area, with initial randomness as explained later. Six trials were performed for each simulation condition.

**Positions of muscle units and hemocytes.** We randomly positioned 900 muscle units in the uneaten state within left and right vertical belts each with the width of 100 μm, locating

200 μm away from each other (Fig 6a, 12 hAPF, yellow particles). These belts imitate two larval muscles. In the area outside the muscle belts, 800 hemocytes were randomly positioned (Fig 6a, 12 h, magenta particles).

**Positions and floating of fat body cells.** Under the conditions in which fat body cells exist from the initial step, 90 fat body cells with constant radii $r_i = R_i$ were randomly positioned in the area outside the muscle belts as in hemocytes (Fig 6b, 12 hAPF, gray particles). In the conditions assuming that fat body cells float later, the initial positions of 70 fat body cells were random throughout the field. To imitate the floating of fat body cell $i$ from the depth, the radius $r_i$ was gradually increased from zero to its maximum radius $R_i$. Assuming a natural pattern, we let $R_i$ range from 18 to 28 μm at equal intervals, with 23 μm in average. The increase in radius was started at 14 hAPF (step 14,400), with the radius increment 0.383 μm every minute; the average-sized cell completed the radius increase in 1 h (Fig 6c, 15 hAPF). If the initial position of fat body cell $i$ overlapped with an uneaten muscle unit within the radius of $R_i$, the step to start increasing $r_i$ was delayed until the uneaten muscle unit is engulfed by a hemocyte or pushed out of the vicinity. This delay avoids the unrealistic situation that fat body cells forcibly float inside the undigested larval muscle because fat body cells themselves likely have no ability to destruct the tissue. We let a fat body cell whose radius is more than zero exhibit active migration, that is, $v_i = 0$ if $r_i = 0$ and $v_i = v_F$ if $r_i > 0$.

**Heterogeneity in hemocyte motility.** To understand the functional significance of heterogeneity in the self-propelling force strength and turning frequency of hemocytes in vivo, we compared three simulation conditions as follows.

- **Homogeneous hemocytes.** The self-propelling force strength $v_{H,i} = 5$ [nN] and the turning probability $p_i = 0.005$ were assigned to all the 800 hemocytes in standard type (Fig 7a).
- **Heterogeneous hemocytes in self-propelling force strength.** A low strength $v_{H,i} = 1$ [nN] was assigned to 400 hemocytes in slow type, whereas a high strength $v_{H,i} = 9$ [nN] was assigned to the other 400 hemocytes in fast type (Fig 7b). The average strength over all the hemocytes became $v_{H,i} = 5$ [nN], which was identical to the homogeneous one.

**Table 1. Simulation parameters.**

| Parameter | Specification | Unit | Value | Equation(s) |
|---|---|---|---|---|
| $\mu$ | friction coefficient | [Ns/m] | † $3.0 \times 10^{-2}$ | (2) |
| $p_i$ | probability to turn a hemocyte or a fat body cell | — | * $5.0 \times 10^{-3}$ | (2) related |
| $v_{H,i}$ | coefficient for the self-propelling force strength of a hemocyte | [N] | *† $5.0 \times 10^{-9}$ | (3) |
| $v_F$ | coefficient for the self-propelling force strength of a fat body cell | [N] | $2.5 \times 10^{-9}$ | (3) |
| $k$ | coefficient for repulsion strength determined by the overlapping length between two particles | [N/m] | $1.0 \times 10^{-3}$ | (4,6) |
| $r_S$ | radius of a muscle unit | [m] | $6.0 \times 10^{-6}$ | (4,6) related |
| $r_H$ | radius of a hemocyte | [m] | $4.0 \times 10^{-6}$ | (4,6) related |
| $r_F$ | radius of a fat body cell | [m] | $1.80 \times 10^{-5}$ to $2.80 \times 10^{-5}$ | (4,6) related |
| $a_{HF}$ | adhesion strength between a hemocyte and a fat body cell | [N] | †§ $2.0 \times 10^{-9}$ | (5) |
| $a_{HH}$ | adhesion strength between two hemocytes | [N] | †§ 0 | (5) |
| $a_{FF}$ | adhesion strength between two fat body cells | [N] | § 0 | (5) |
| $a_E$ | adhesion strength between a muscle unit and a hemocyte, representing phagocytosis | [N] | $1.0 \times 10^{-8}$ | (5) |

* Changed to two-grouped constants in the heterogeneity simulations. See Sect 2.2.3. † Changed over time in the stabilization simulations. See Sect 2.2.4. § Changed in supplementary simulations (S6 Fig).

The common turning probability $p_i = 0.005$ was assigned to all the hemocytes as in the homogeneous case.

- **Heterogeneous hemocytes in turning frequency.** A frequent-turning probability $p_i = 0.009$ was assigned to 400 hemocytes in meander type, and an infrequent-turning probability $p_i = 0.001$ was assigned to the other 400 hemocytes in straight type (Fig 7c). The average turning probability over all the hemocytes became $p_i = 0.005$, which was identical to the homogeneous one. The common self-propelling force strength $v_{\text{H},i} = 5$ [nN] was assigned to all the hemocytes as in the homogeneous case.

**Stabilization.** To understand the mechanism of how muscle units reduce their moving speed in vivo, we performed five types of stabilization operations to examine if these cause arrangement changes similar to that in vivo. The relevant parameters were linearly changed during the period of steps 129,600–216,000 (30–50 hAPF) at intervals of 120 steps (1 min) to eventually decrease the average speed of muscle units by 80%. To what extent we change each parameter was determined via linear regression; the explanatory variable on x-axis was the extents of several parameter changes, whereas the response variable on y-axis was the resultant percentages of the average muscle unit speed at 30 hAPF (step 129,600) to that at 50 hAPF (step 273,600) (S7 Fig). The x value at which y took 80% on the regression line was applied to the parameter change in each operation.

- **Spatial confinement of the simulation field.** The thickness of the rectangles on the top and bottom each was increased from 0 to each 90 μm during the period (Fig 8a, colored dark blue). Particle $i$ (muscle unit, hemocyte, or fat body cell) in contact with the rectangle additionally receives a repulsion force as below:

$$\mathbf{f}_{i,\text{wall}}(t) = -k(r_i + w(t) - y_i(t))\mathbf{e}_{i,\text{wall}}, \tag{6}$$

where $w(t)$ is the vertical thickness of the rectangle, $y_i(t)$ is the vertical distance from $\mathbf{x}_i(t)$ to the nearest top or bottom edge of the simulation field, and $\mathbf{e}_{i,\text{wall}}$ is a unit vector in the perpendicular direction to the nearest top or bottom edge. Accordingly, the periodic boundary was retained on the left and right edges but lost on the top and bottom edges during the spatial confinement.
- **Decrease in the self-propelling force strength of hemocytes.** The strength $v_{\text{H},i}(t)$ was decreased to 0.80 times during the period (Fig 8b).
- **Increase in the friction coefficient of all the particles.** The value $\mu(t)$ was increased from 0.030 to 0.039 Ns/m during the period (Fig 8c). We assume a change in a property of the environment such as the density of the extracellular matrix, which affects the moving speed of all the particles.
- **Increase in the adhesion strength between a hemocyte and a fat body cell.** The value $a_{\text{HF}}(t)$ was increased from 2.0 to 4.0 nN during the period (Fig 8d).
- **Increase in the adhesion strength between two hemocytes.** The value $a_{\text{HH}}(t)$ was increased from 0 to 2.2 nN during the period (Fig 8e).

## 4.6. In silico evaluation

To evaluate to what extent simulated muscle units are stable, the moving speed of muscle unit coordinates was calculated over time. The moving distance per 6 min (720 steps) was divided by six to obtain the speed per minute. How efficiently muscle units spread from the left and right larval muscles in the simulation field (Fig 6a and 6b, 12 h) was evaluated based

on the sample variance of the coordinates of muscle units. At each time point, the variance along the horizontal axis in the left half and that in the right half were averaged to calculate the common variance, termed 'horizontal variance.'

To evaluate to what extent the simulated arrangements of muscle units resemble the in vivo arrangement, the number of nearby muscle units and the ring radius at some time points were calculated using the same methods to those in vivo (Sect 4.3). A single muscle unit in silico was interpreted as a single sarcolyte in vivo. The central coordinates of muscle units in the whole simulation field were used for calculation without consideration of the periodic boundary. The dataset of each measurement is stored in S1 Dataset.

## Supporting information

**S1 Fig. Acquisition of signal intensity difference images in dummy data.** Let original images consist of two frames (left), from which one signal intensity difference image are obtained (right). The middle shows the semi-transparent overlay of the two original images to help understanding. Assume white particles are moving rightward with different speeds. The slower a particle moves, the smaller the area of signal intensity-changed pixels (colored white in the right images); particle #1 yields the smallest signal intensity difference, and #2 yields the second smallest. A moment with many #1-like particles outputs a smaller value of $\Delta F(t)$, averaged signal intensity difference, than that with many #2-like particles. Signal intensity difference sometimes underestimates the speed: e.g., particles #3 and #4 with different speeds produce the same area of signal intensity difference because both move to areas that exclude the previous areas; the middle particle of #5 is apparently vanished in the signal intensity difference images due to nearby particles.
(PDF)

**S2 Fig. In vivo signal intensity data to calculate the normalized signal intensity difference.** Left: average signal intensity of the original images ($F_0(t)$), right: average signal intensity of the signal intensity difference images before normalization ($\Delta F(t)$). The normalized signal intensity difference ($\Delta F(t)/F_0(t)$) is shown in Fig 1c with corresponding colors indicating three individuals.
(PDF)

**S3 Fig. Algorithm of persistent homology in dummy data.** A 'ring-like' structure is recognizable in the left coordinate data but not in the right. This subjective assessment of 'ring-likeness' corresponds to the length of the life time—the subtraction of the birth time (roughly equivalent to half the average inter-point distance enclosing the ring) from the death time (roughly equivalent to the radius of the ring). In the actual coordinate data, a number of such structures can be detected throughout the field, generating multiple subsets of birth, death, and life times.
(PDF)

**S4 Fig. Persistence diagram of in vivo coordinate data.** Left: 24 hAPF, right: 40 hAPF. The 'ring-like' structures based on persistent homology are annotated by blue circles with >10 μm life time, a threshold indicated by the blue dotted lines. The death time of the 'ring-like' structures is defined as the ring radius, whose histogram is shown in Fig 2c.
(PDF)

**S5 Fig. Detail of the speed and steering consistency of hemocytes.** (a) Trajectories of all the analyzed hemocytes. The initial positions are aligned to the origin. The color code corresponds to the rank of total migratory trajectory length: red, short; blue, long. (b) Heatmap of

instantaneous speed for individual hemocytes. Time series of instantaneous speed for individual cells are aligned horizontally. (c) A representative trajectory for a hemocyte. Color code indicates time. Circular plot indicates angular difference in moving orientation from a frame before at each time point. 0 deg to the right, 180 deg to the left, rightward to the top, and leftward to the bottom. (d) Heatmap of difference in moving orientation from a frame before at each time point for individual hemocytes. Time series of $\cos\theta_i(n)$ (defined in Sect 2.1.4) for individual cells are aligned horizontally.
(PDF)

**S6 Fig. Early-stage simulations with different adhesion strength.** Snapshots are captured at 21 hAPF (step 64,800). Top left: no adhesion, $a_{HF} = a_{HH} = a_{FF} = 0$. Top right: adhesion between a hemocyte and a fat body cell, $a_{HF} = 2.0$ [nN] and $a_{HH} = a_{FF} = 0$; these parameters are applied in the main simulations. Bottom left: adhesion between hemocytes, $a_{HH} = 2.0$ [nN] and $a_{HF} = a_{FF} = 0$. Bottom right: adhesion between fat body cells, $a_{FF} = 2.0$ [nN] and $a_{HF} = a_{HH} = 0$. Lines below the snapshots show the plot legend for the speed (top plot) and the horizontal variance (bottom plot). The plots are shown as in Fig 6d and 6e. In all the conditions, fat body cells start floating after 14 hAPF (shaded in the plots), and hemocytes are homogeneous in motility. See S6 Video for the simulation video.
(PDF)

**S7 Fig. Linear regression for stabilization operations.** For each operation, we perform simulations in which a parameter is gradually changed by a degree (explanatory variable on the x-axis) and obtain the percentage of the average speed of muscle units before the change to that after the change (Response variable on the y-axis). Each circle represents the measurement from a simulation trial. A series of simulations yields a regression line, by which we determine a parameter change that reduces the muscle unit speed by 80% (arrow).
(PDF)

**S1 Video. In vivo imaging of sarcolytes**. The upper row indicates the original images, while the lower row indicates the signal intensity difference images, with three individuals along the columns. The brightness is increased with a common degree to improve visibility. Time after puparium formation (hAPF) is shown at the top left (h:min). Snapshots of the individual #1 are shown in Fig 1a and 1b. Black, orange, and blue curves in Fig 1c represent the individuals #1, #2, and #3, respectively.
(PDF)

**S2 Video. In vivo imaging of fat body cells.** Time after puparium formation (hAPF) is shown at the bottom right (h:min:s). Snapshots are shown in Fig 3a.
(PDF)

**S3 Video. Simulations related to the existence of fat body cells.** The numbers on the top left indicates the step and its corresponding time (hAPF). Snapshots are shown in Fig 6a–6c.
(PDF)

**S4 Video. Simulations related to the micro-heterogeneity of hemocytes.** The numbers on the top left indicates the step and its corresponding time (hAPF). Conditions correspond to Fig 7a–7c. In the heterogeneous cases, hemocytes are colored blue in the slow or meander type and red in the fast or straight type.
(PDF)

**S5 Video. Simulations related to stabilization operations.** The numbers on the top left indicates the step and its corresponding time (hAPF). Snapshots are shown in Fig 8a–8e.
(PDF)

**S6 Video. Simulations related to adhesion strength.** The numbers on the top left indicates the step and its corresponding time (hAPF). Snapshots are shown in S6 Fig.
(PDF)

**S1 Dataset. In vivo and in silico dataset.** The ZIP file contains tab-separated text files of data used for the main plots.
(PDF)

**S1 File. Simulation codes.** The ZIP file contains the source codes in C++: "main.cpp," "Monitor.cpp," and "Monitor.hpp" with a "makefile" file.
(ZIP)

## Acknowledgments

We thank Dr. Naoko Kaneko (Doshisha University), Dr. Yuichiro Sueoka (The University of Osaka), Dr. Yuriko Sobu (Doshisha University), Dr. Taishi Mikami (Kajima Corporation), Dr. Yasuaki Hiraoka (Kyoto University), and Dr. Hiraku Nishimori (Meiji University) for fruitful discussions. We thank Dr. Masahiro Shimizu (Nagahama Institute of Bio-Science and Technology) for providing part of the simulation codes.

## Author contributions

**Conceptualization:** Daiki Umetsu, Takeshi Kano.

**Data curation:** Daiki Wakita, Daiki Umetsu.

**Formal analysis:** Daiki Wakita, Satoshi Yamaji, Daiki Umetsu.

**Funding acquisition:** Daiki Umetsu, Takeshi Kano.

**Investigation:** Daiki Umetsu.

**Methodology:** Daiki Wakita, Satoshi Yamaji.

**Project administration:** Takeshi Kano.

**Resources:** Daiki Umetsu.

**Software:** Daiki Wakita.

**Supervision:** Takeshi Kano.

**Validation:** Daiki Wakita.

**Visualization:** Daiki Wakita, Daiki Umetsu.

**Writing – original draft:** Daiki Wakita, Daiki Umetsu.

**Writing – review & editing:** Takeshi Kano.

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
