## [Decision Letter · Decision Letter 0]

6 Jun 2025

PCOMPBIOL-D-25-00709

Dual-purpose dynamics emerge from a heterogeneous cell population in *Drosophila* metamorphosis

PLOS Computational Biology

Dear Dr. Kano,

Thank you for submitting your manuscript to PLOS Computational Biology. After careful consideration, we feel that it has merit but does not fully meet PLOS Computational Biology's publication criteria as it currently stands. Therefore, we invite you to submit a revised version of the manuscript that addresses the points raised during the review process.

Please submit your revised manuscript within 60 days Aug 06 2025 11:59PM. If you will need more time than this to complete your revisions, please reply to this message or contact the journal office at ploscompbiol@plos.org. Please include the following items when submitting your revised manuscript:

We look forward to receiving your revised manuscript.

Kind regards,

Jing Chen

Academic Editor

PLOS Computational Biology

Pedro Mendes

Section Editor

PLOS Computational Biology

**Additional Editor Comments:**

Both reviewers have complimented on the overall significance and rigor of the work. As suggested by one of the reviewers, it is important to provide more testable predictions to strengthen the work.

**Journal Requirements:**

Please amend your detailed Financial Disclosure statement. This is published with the article. It must therefore be completed in full sentences and contain the exact wording you wish to be published. State what role the funders took in the study. If the funders had no role in your study, please state: "The funders had no role in study design, data collection and analysis, decision to publish, or preparation of the manuscript.".

**Reviewers' comments:**

Reviewer's Responses to Questions

**Comments to the Authors:**

Reviewer #1: This paper reports an integrated in vivo imaging and agent-based modeling study on how cellular heterogeneity governs the collective behavior during muscle remodeling in Drosophila pupae during metamorphosis. The authors find the sarcolyte movement slows down over time and becomes more spatially uniform. They propose that macro-heterogeneity (across hemocytes and fat body cells) and micro-heterogeneity (within the same type, e.g., variability in hemocyte motility) contribute to both rapid dispersal and stable pattern formation, an interesting phenomenon they term “dual-purpose dynamics.” Their agent-based model shows that delayed appearance of fat body cells facilitates early spread of sarcolytes followed by spatial stabilization, and variation in hemocyte motility allows dispersal while keeping sarcolytes locally stable. In addition, spatial confinement reproduces the slowing and pattern formation in the later stage.

Overall, the paper is very well written. In vivo imaging data are from multiple individuals and developmental time points with consistent trends and robustness. Appropriate statistical tests are used to quantify the changes in motion and spatial patterns. The agent-based model includes multiple cell types, differential adhesion, and motility heterogeneity (turning frequency), with biologically motivated rules and parameters.

The main complaint I have is the lack of model to experiment feedback – the model predictions do not inform more experiments to validate the discovery. The study would be significantly strengthened by an additional perturbation experiment, e.g., delay fat body emergence or modify hemocyte heterogeneity, to confirm causal relationship.

I also found the presentation of the data can be improved.

Figure 1 (b): the visual distinction between static and moving sarcolytes in the difference images are not immediately intuitive. A color-coded image may be easier to understand than the grayscale differences. (c) is missing a legend. Can the signal intensity difference in Fig 1c be converted to speed and be compared with modeling result in e.g., Fig 8f?

Figure 2 (a): overlaying sarcolyte coordinates from 6 individuals makes spatial trends difficult to discern. It might be a good idea to color-code individuals.

Figure 4 (a): Cumulative displacement (panel a) is not the best format for comparing individual variability. Individual trajectory maps or vector fields would be more effective.

Figure 5. Adding a flowchart of model dynamics could help readers less familiar with agent-based modeling.

Figure 6. how many replicate simulations were done for each parameter set? Do we not expect error bars for speed and spatial variance measurements?

Figuer 8. Not clear what error bars are, and if the differences are significant.

Table 1: adding another column indicating if the model parameter values are estimated, measured, or fitted would be better.

S1 Fig – I find it very confusing.

S3 Fig – standard TDA illustration, maybe no need to include

Reviewer #2: The study presents computational results that complement their in vivo observations of muscle remodeling in Drosophila, offering insight into how cellular heterogeneity shapes collective behavior during development. The authors conceptualize the remodeling process as a dynamic interaction between migrating hemocytes, fat body cells, and decomposing muscle fragments (sarcolytes), and develop a mathematical model to probe the causal mechanisms underlying sarcolyte dynamics. They introduce computational tools to evaluate how macro-heterogeneity (across cell types) and micro-heterogeneity (within hemocyte behavior) contribute to the emergent spatial organization of sarcolytes, revealing a dual-purpose mechanism by which biological systems achieve both rapid dispersal and patterned arrangement.

The figures, methods, and explanations collectively support the idea that the developed computational model is a reasonable representation of the empirical system. Overall, the references and detailed results are consistent and well-explained. The codes are included and well documented within the included material links.

Major Comments/Issues

-None

Minor Comments/ Issue

1. Grammar correction (Author Summary)

In the author summary the sentence: “We then used mathematical tools to find what affect the

movement of muscle pieces.”

The word affects should replace the word affect.

2. Figure caption and terminology order (Section 2.1.1)

(line 72) Section 2.1.1 introduces hours after puparium formation (hAPF), the figures referenced in the draft appear before this introduction of hAPF.

While minor, it would improve readability to introduce hAPF before its first usage in the figures.

3. Definition of "network-like arrangement" (Section 2.1.2)

The definition of a somewhat network like arrangement is not clear from the provided images, either providing a colored panel highlighting this in the images and/or a definition such as network-like arrangement typically refers to a spatial organization of cells or cellular structures that resembles a mesh, web, or interconnected lattice.

Even if the term is familiar within the field, a definition or citation would strengthen the text.

4. Force representation clarification (Line 573)

The force repulsion term:

(-k(ri+rj-Xj(t)-Xi(t)|)

This gives the repulsion terms between different particles, this might need more expansion in the details of the justification of why add the two radii sizes in the repulsion representation. Is this meant to be an approximation given along with the unit vector between the two coordinate points of the circumference area of one particle directed at another?

The primary focus for the force seems to be the aij term, but the overall strength of the work would benefit if the repulsive terms were elaborated upon.

5. Radius growth clarification (Page 27, lines 617–620)

The statement about fat body cell growth isn't entirely clear. At a growth rate of 0.45 μm per minute, the radius would increase by 27 μm over one hour, which exceeds the stated average radius of 23 μm.

Consider rephrasing this section for clarity

6. Follow-up on fat body cell growth modeling

Building on Comment 5:

Is there any plan to explore more biologically motivated growth dynamics, such as varying the fat body cell radius based on spatial constraints, relative location, or developmental timing, rather than applying a constant rate of increase?

Incorporating such variation might better reflect biological heterogeneity and could yield further insights into future work.

**Have the authors made all data and (if applicable) computational code underlying the findings in their manuscript fully available?**

Reviewer #1: Yes

Reviewer #2: Yes

PLOS authors have the option to publish the peer review history of their article (what does this mean?). If published, this will include your full peer review and any attached files.

Reviewer #1: No

Reviewer #2: No

**Figure resubmission:**
---

## [Decision Letter · Decision Letter 1]

14 Jul 2025

Dear Dr. Kano,

We are pleased to inform you that your manuscript 'Dual-purpose dynamics emerge from a heterogeneous cell population in *Drosophila* metamorphosis' has been provisionally accepted for publication in PLOS Computational Biology.

Best regards,

Jing Chen

Academic Editor

PLOS Computational Biology

Pedro Mendes

Section Editor

PLOS Computational Biology

Reviewer's Responses to Questions

**Comments to the Authors:**

Reviewer #2: Overall, the updated sections are consistent, well integrated, and clearly explained. The authors have effectively addressed the reviewer comments, which has strengthened the main computational findings and improved the connection to relevant empirical studies. The revisions enhance the clarity and impact of the work, and the key concerns raised in the initial review have been satisfactorily resolved.

**Have the authors made all data and (if applicable) computational code underlying the findings in their manuscript fully available?**

Reviewer #2: Yes

PLOS authors have the option to publish the peer review history of their article (what does this mean?). If published, this will include your full peer review and any attached files.

Reviewer #2: No

---

## [Editor Report · Acceptance letter]

PCOMPBIOL-D-25-00709R1

Dual-purpose dynamics emerge from a heterogeneous cell population in *Drosophila* metamorphosis

Dear Dr Kano,

I am pleased to inform you that your manuscript has been formally accepted for publication in PLOS Computational Biology. Your manuscript is now with our production department and you will be notified of the publication date in due course.

With kind regards,

Anita Estes
